# Tri-methylation of histone H3 lysine 4 facilitates gene expression in ageing cells

Cristina Cruz[†], Monica Della Rosa[†], Christel Krueger, Qian Gao[‡], Dorottya Horkai, Michelle King, Lucy Field[§], Jonathan Houseley*

Epigenetics Programme, The Babraham Institute, Cambridge, United Kingdom

**Abstract** Transcription of protein coding genes is accompanied by recruitment of COMPASS to promoter-proximal chromatin, which methylates histone H3 lysine 4 (H3K4) to form H3K4me1, H3K4me2 and H3K4me3. Here, we determine the importance of COMPASS in maintaining gene expression across lifespan in budding yeast. We find that COMPASS mutations reduce replicative lifespan and cause expression defects in almost 500 genes. Although H3K4 methylation is reported to act primarily in gene repression, particularly in yeast, repressive functions are progressively lost with age while hundreds of genes become dependent on H3K4me3 for full expression. Basal and inducible expression of these genes is also impaired in young cells lacking COMPASS components Swd1 or Spp1. Gene induction during ageing is associated with increasing promoter H3K4me3, but H3K4me3 also accumulates in non-promoter regions and the ribosomal DNA. Our results provide clear evidence that H3K4me3 is required to maintain normal expression of many genes across organismal lifespan.
DOI: https://doi.org/10.7554/eLife.34081.001

*For correspondence:
jon.houseley@babraham.ac.uk

[†]These authors contributed equally to this work

Present address: [‡]Adaptimmune Ltd, Abingdon, United Kingdom; [§]Molecular Haematology Unit, The MRC Weatherall Institute of Molecular Medicine, Oxford, United Kingdom

Competing interests: The authors declare that no competing interests exist.

## Introduction

H3K4me3 is ubiquitously observed on nucleosomes at the 5' end of eukaryotic genes undergoing active transcription by RNA polymerase II (see for example [*Barski et al., 2007*; *Bernstein et al., 2005*; *Liu et al., 2005*; *Zhang et al., 2009*]). The tight correlation between promoter H3K4me3 and transcriptional activity has led to H3K4me3 being widely considered as an activating mark, however, direct evidence for a role in steady-state transcription or gene induction is rare and controversial (reviewed in [*Howe et al., 2017*]).

H3K4me3 is primarily deposited by the highly conserved COMPASS complexes (*Briggs et al., 2001*; *Krogan et al., 2002*; *Miller et al., 2001*; *Roguev et al., 2001*; *Santos-Rosa et al., 2002*). Budding yeast has a single COMPASS complex containing the catalytic SET-domain protein Set1 and core structural proteins Swd1 and Swd3, along with Sdc1, Bre2, Swd2 and Spp1 (*Figure 1A*). Set1, Swd1 and Swd3 are required for all H3K4 methylation activity, while Sdc1 and Bre2 are necessary for di- and tri-methylation (*Dehé et al., 2006*; *Schneider et al., 2005*). Neither Swd2 nor Spp1 are required for H3K4 methylation in vitro (*Takahashi et al., 2011*), but mutation of Swd2 almost completely abrogates H3K4me3 and reduces H3K4me2 in vivo (*Cheng et al., 2004*; *Lee et al., 2007*), while loss of Spp1 dramatically reduces H3K4me3 in vivo without impacting H3K4me1/2 (*Dehé et al., 2006*; *Schneider et al., 2005*).

Promoter recruitment of COMPASS starts with the RNA pol II elongation factors Paf and FACT, which activate the H2B-ubiquitylation activity of Rad6/Bre1 (*Pavri et al., 2006*); this activity, and a non-mutated H2B[123K] substrate, are vital for COMPASS recruitment and di-/tri-methylation (*Dover et al., 2002*; *Krogan et al., 2003*; *Sun and Allis, 2002*; *Wood et al., 2003*). Activation of Rad6/Bre1 and COMPASS recruitment also require interactions with the Ser-5 phosphorylated CTD of RNA polymerase II, providing a tight transcriptional connection (*Krogan et al., 2003*; *Ng et al., 2003*; *Xiao et al., 2005*). COMPASS recruitment then occurs through interactions of Swd2 with

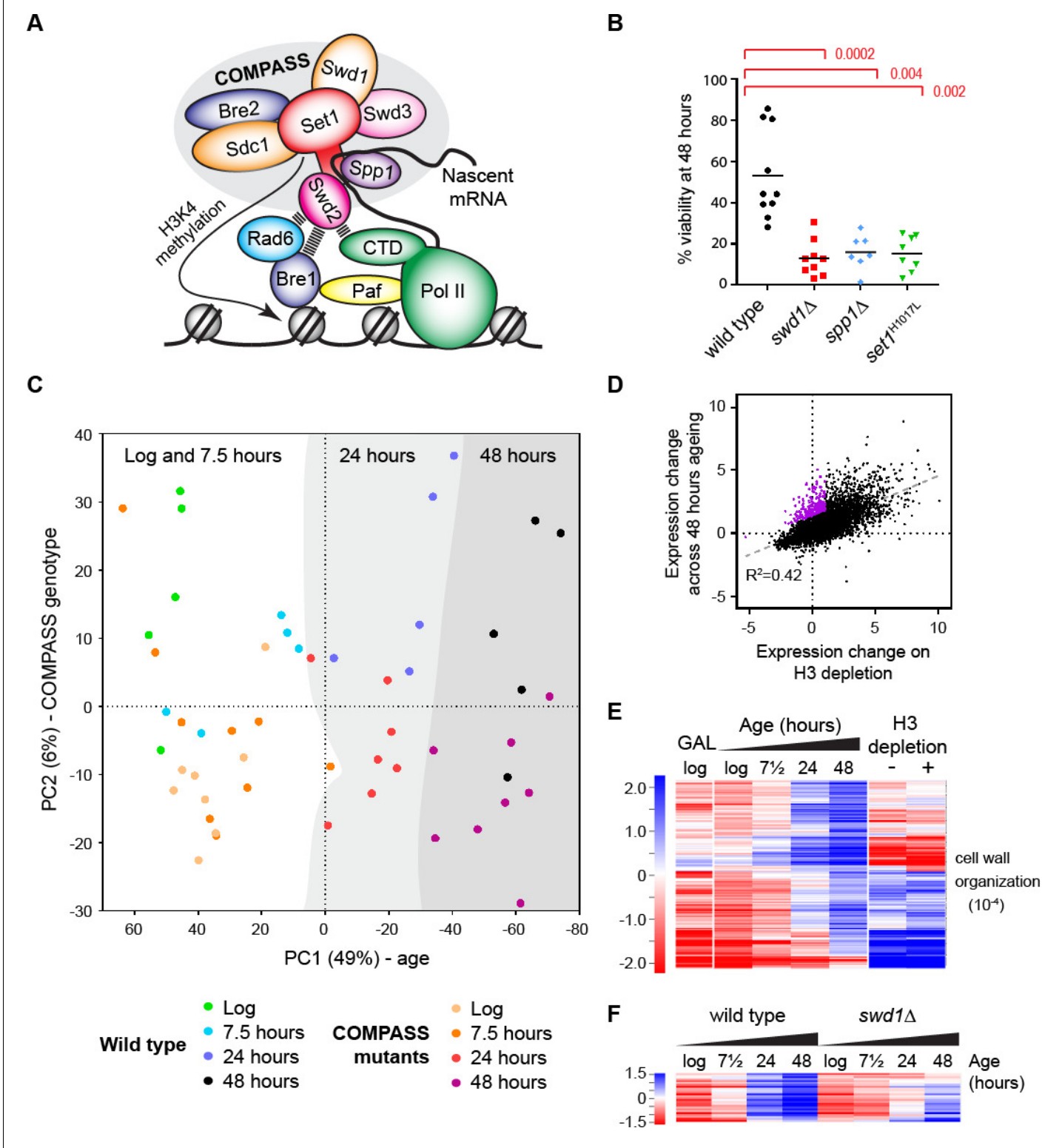

**Figure 1.** The ageing transcriptome in wild type and COMPASS mutants. (**A**) Schematic of COMPASS recruitment for deposition of H3K4me3. (**B**) Viability of mother cells after 48 hr in liquid YPD culture determined using the MEP system. p-values calculated by Kruskal-Wallis test, n = 10 (wild type), n = 9 (*swd1Δ*), n = 7 (*spp1Δ*), n = 8 (*set1*^H1017L). (**C**) PCA plot of protein-coding gene expression for the 52 mRNAseq libraries generated from wild type and COMPASS mutants across 48 hr ageing. COMPASS mutants have been shown in the same colours for simplicity and to reflect their pooling in the initial DESeq2 analyses (*Figure 2A,B*). Grey stripes define clusters of samples of different ages, showing that wild type and COMPASS mutants are

*Figure 1 continued*

similarly distributed on PC1, which primarily reflects age. (D) Plot of $\log_2$-transformed gene expression change from log phase to 48 hr ageing versus before to after H3 depletion. Genes that increase >2 fold more than average during ageing but increase <2 fold on H3 depletion are highlighted in purple. (E) Hierarchical clustering analysis of $\log_2$-transformed protein-coding mRNA change for genes differentially expressed between log phase and 48 hr ageing in wild type but not on H3 depletion or between glucose and galactose media. GO enrichment assigned using GOrilla as in *Figure 1—figure supplement 2A*, the order of magnitude of the FDR-corrected p-value is quoted in brackets. (F) Hierarchical clustering analysis across ageing for wild type and *swd1Δ* of the subset of genes from E that are significantly differentially expressed between wild-type and *swd1Δ* cells at 48 hr (DESeq2 p<0.01 n = 5 wild type, n = 4 *swd1Δ*).

DOI: https://doi.org/10.7554/eLife.34081.002

The following figure supplements are available for figure 1:

**Figure supplement 1.** Supplement to the ageing transcriptome in wild type and COMPASS mutants.
DOI: https://doi.org/10.7554/eLife.34081.003
**Figure supplement 2.** Age-linked changes in gene expression profile.
DOI: https://doi.org/10.7554/eLife.34081.004

active H2B-associated Rad6/Bre1, and of Set1/Spp1 with nascent mRNA (*Battaglia et al., 2017*; *Luciano et al., 2017*; *Sayou et al., 2017*; *Thornton et al., 2014*). In mammals, additional CxxC domains in Cfp1 (the orthologue of yeast Spp1) and some SET-domain proteins further specify and diversify COMPASS activity, allowing DNA methylation-dependent activity and selective recruitment of COMPASS to promoter and enhancer sequences (*Brown et al., 2017*; *Clouaire et al., 2012*; *Hu et al., 2017*; *Thomson et al., 2010*).

The importance of H3K4me3 in transcription has been difficult to study as loss of Set1 also abrogates H3K4me1 and H3K4me2. Even so, loss of Set1 does not drastically impair transcription in yeast, and the extent to which subsets of genes are mis-regulated in *set1Δ* cells has been controversial with later studies reporting mostly gene upregulation (*Guillemette et al., 2011*; *Margaritis et al., 2012*; *Miller et al., 2001*; *Santos-Rosa et al., 2002*; *Venkatasubrahmanyam et al., 2007*; *Weiner et al., 2012*). This suggests a primarily repressive role for H3K4 methylation, particularly evident for ribosome synthesis factors (*Weiner et al., 2012*), noncoding RNA-regulated genes (*Berretta et al., 2008*; *Camblong et al., 2009*; *Pinskaya et al., 2009*) and heterochromatic regions (*Briggs et al., 2001*; *Li et al., 2006*; *Nislow et al., 1997*; *Venkatasubrahmanyam et al., 2007*). To determine the importance of H3K4me3 specifically, studies in yeast have focused on cells lacking Spp1 which is required for most (but not all) H3K4me3, and have observed virtually no impact on steady state gene expression or on acute gene induction during media shifts (*Howe et al., 2017*; *Margaritis et al., 2012*). In contrast, loss of CFP1 in mammalian cells has concrete effects on gene expression, causing both up- and down-regulation of many genes in mouse ES cells and reduced transcription in oocytes (*Brown et al., 2017*; *Yu et al., 2017*). Interpretation of these observations is complicated by the additional binding modes of CFP1 that target COMPASS to promoters and enhancers, so loss of H3K4me3 may not be the only effect of CFP1 depletion (*Brown et al., 2017*; *Hu et al., 2017*). One specific report ties H3K4me3 mechanistically to activation of p53-regulated genes in response to DNA damage through recruitment of TAF3 in HCT116 cells, however this may be cell-type specific as the same response was not impaired in mouse ES cells lacking CFP1 (*Clouaire et al., 2014*; *Lauberth et al., 2013*). Overall, H3K4 methylation has the potential to both repress and activate transcription, but a general role for H3K4me3 in transcriptional activation remains unproven.

An association between histone modifications and ageing is well recognised (reviewed in [*Maleszewska et al., 2016*]). For example, defects in H3K4, H3K36 and H3K27 methylation all affect lifespan presumably through impacts on gene expression, though in most cases the downstream expression differences remain obscure (*Alvares et al., 2014*; *Greer et al., 2010*; *Li et al., 2010*; *Maures et al., 2011*; *McColl et al., 2008*; *Sen et al., 2015*; *Siebold et al., 2010*). Remarkably, a gross genome-wide loss of histones has been reported to accompany ageing in yeast and cultured human cells (*Feser et al., 2010*; *Ivanov et al., 2013*; *O'Sullivan et al., 2010*), leading to an opening of chromatin structure particularly at repressed promoters and causing widespread induction of normally repressed genes (*Hu et al., 2014*). General opening of chromatin structure has therefore been proposed to underlie many age-linked gene expression differences (*Hu et al., 2014*), and it is easy to imagine how alterations in canonical transcription-associated histone modifications could

exacerbate these phenotypes. However, the impacts of histone modification defects on particular genes as cells age remain largely unknown due to the difficulty of de-convolving these from the massive impact of the ageing process itself.

Here we analyse the importance of H3K4me3 in facilitating gene expression across the lifetime of budding yeast, and find that H3K4me3 is critical for the full expression of many genes that are induced with age. This contrasts with and is separable from the well-characterised repressive function of H3K4 methylation, which we show declines with age. We validate the importance of H3K4 trimethylation in the inducible expression of a subset of genes, demonstrating a direct role for H3K4me3 in maintaining normal expression of many genes across organismal lifespan.

## Results

### A transcriptomic dataset for ageing wild type and COMPASS mutants

A published micromanipulation screen of 264 yeast mutants found that cells lacking COMPASS components Swd1 or Swd3 have replicative lifespans ~ 20% shorter than wild type (*Smith et al., 2008*). Replicative lifespan measures the number of times a mother cell can divide before losing viability, compared to chronological lifespan which measures the time for which cells remain viable irrespective of cell division. To determine the importance of COMPASS in controlling gene expression and chromatin structure across replicative lifespan we introduced selected COMPASS mutants into the Mother Enrichment Program (MEP) background; the MEP facilitates isolation of highly aged yeast in sufficient quantities for standard molecular techniques, and provides simple comparative methods to assess lifespan (*Lindstrom and Gottschling, 2009*). The lifespan methods, although less precise than microdissection or microfluidics, are performed in the same growth conditions as for other molecular techniques and therefore allow integration of lifespan and molecular data. As expected, MEP *swd1Δ* cells show a substantial and significant defect in replicative lifespan, being only ~50% viable at 24 hr and almost completely inviable after 48 hr in YPD (*Figure 1—figure supplement 1A*). Importantly, this lifespan defect was recapitulated in catalytic dead *set1$^{H1017L}$* and H3K4me3-defective *spp1Δ* mutants, showing that the lifespan defect results specifically from loss of H3K4me3 (*Figure 1B* and *Figure 1—figure supplement 1B*).

As COMPASS has non-histone substrates (*Zhang et al., 2005*), we were concerned that this viability difference may not be attributable to H3K4 methylation. Introducing histone point mutations into the MEP background proved challenging, but we were able to create MEP haploids of H3$^{K4A}$ and H3$^{K4R}$. Selection against daughter cells is imperfect in haploid MEPs (*Lindstrom and Gottschling, 2009*), a problem that can be partially offset by measuring viability at 24 hr, and at this point we observed that viability of the mutants was substantially and significantly reduced compared to the wild-type (*Figure 1—figure supplement 1C*). COMPASS mutants also have a reduced chronological lifespan due to stimulation of apoptosis by the H3K79 methyltransferase Dot1 (*Walter et al., 2014*); however no suppression of the replicative lifespan defect was observed in *swd1Δ dot1Δ* double mutants (*Figure 1—figure supplement 1D*) so the replicative and chronological lifespan defects cannot be attributed to the same mechanism. Overall, although *spp1Δ* mutants have a very weak phenotype during normal growth, this mutation dramatically impairs lifespan as do all tested mutants that impact H3K4 methylation, indicating that H3K4me3 has a prominent function in ageing cells.

To understand the gene expression differences between wild-type and COMPASS mutants, we obtained mRNAseq profiles from multiple replicates of wild-type, *swd1Δ*, *set1$^{H1017L}$* and *spp1Δ* cells in log phase and aged for 7.5, 24 or 48 hr in YPD, forming a dataset of 52 libraries (*Table 1*). Principle component (PC) analysis of this dataset shows that PC1, 49% of variance, corresponds to age with both wild type and COMPASS mutant samples of equivalent ages having similar PC1 values (*Figure 1C*), although the log and 7.5 hr samples are similar and do not segregate well. PC2, 6% of variance, broadly segregates wild type from COMPASS mutant libraries (*Figure 1C*, note that wild type samples in blue/green lie towards the top whilst COMPASS mutants in yellow/red are largely in the lower half). The small effect of COMPASS genotype is not unexpected given previous reports, but shows that the gene expression phenotype caused by COMPASS mutation is separable from the gene expression phenotype caused by ageing. This gives us confidence that gene expression

**Table 1.** mRNAseq libraries analysed in this work.

| Genotype | Log phase | 7.5 hr | 24 hr | 48 hr |
|---|---|---|---|---|
| wild type | 5 | 5 | 5 | 5 |
| swd1Δ | 4 | 3 | 3 | 4 |
| set1$^{H1017L}$ | 2 | 3 | 2 | 2 |
| spp1Δ | 2 (3 more added in *Figure 2C*) | 3 | 2 | 2 (3 more added in *Figure 2C*) |

DOI: https://doi.org/10.7554/eLife.34081.005

differences at equivalent times do not simply represent faster or slower ageing, which may otherwise be suggested based on the truncated lifespan of the COMPASS mutants.

Differential expression analysis of the wild-type dataset identified 2842 out of 6662 annotated open reading frames (ORFs) that differ significantly between log and 48 hr, with very similar results in the *swd1Δ* mutant (*Figure 1—figure supplement 2A*). GO analysis of these ORFs confirmed many previous findings: translation-associated terms are down-regulated relative to average with age (*Hu et al., 2014*; *Janssens et al., 2015*; *Kamei et al., 2014*; *Yiu et al., 2008*), while *cell wall organisation*, *hexose transport*, *sporulation*, *tricarboxylic acid cycle* and *DNA integration* (i.e. transposon activity) are upregulated as variously reported (*Hu et al., 2014*; *Kamei et al., 2014*; *Koc et al., 2004*; *Lesur and Campbell, 2004*). Genes upregulated with age are generally expressed at low levels in young cells, while genes that are highly expressed in young cells tend to be down-regulated with age relative to average as previously observed (*Figure 1—figure supplement 2B*) (*Hu et al., 2014*); in absolute terms, it has been shown that all yeast genes are actually induced to a greater or lesser extent during ageing, and we therefore refer to all gene expression changes as relative to average (*Hu et al., 2014*).

Age-related gene induction has been directly attributed to loss of histones, and we observe a strong correlation between age-linked gene expression and previously described changes following histone H3 depletion (*Figure 1D*)(*Gossett and Lieb, 2012*; *Hu et al., 2014*). We were interested to know if any particular category of genes is upregulated with age but not histone depletion, and so filtered for genes that are upregulated 2-fold more than average with age but increase less than 2-fold on H3 depletion (*Figure 1D* purple). We also filtered out genes repressed by the galactose to glucose shift used for H3 depletion in the Gossett and Lieb dataset, as the effect of H3 depletion for these genes is not determined. This left a core set of 204 genes, enriched for *cell wall organization* functions, that are robustly upregulated during ageing but not on H3 depletion (*Figure 1E*). This demonstrates that candidate age-linked gene expression programmes can be identified in yeast. Remarkably, 13% of these genes are significantly under-expressed in the *swd1Δ* mutant at 48 hr but not at log phase (*Figure 1F*), and also under-expressed in *set1*$^{H1017L}$ and *spp1Δ* mutants (data not shown), suggesting that cryptic functions for H3K4 methylation in facilitating gene expression are unmasked during the widespread gene induction that accompanies ageing.

## Requirement for H3K4me3 to facilitate expression of genes induced with age

The small subset of genes induced during ageing but not histone depletion is highly restricted and cannot be extrapolated to infer general effects of H3K4 methylation on gene expression. We therefore performed a differential expression analysis to identify H3K4 methylation-dependent genes at different ages across all genes instead of only those unaffected by histone depletion. Compared to gene expression changes that occur during ageing the differences between wild type and COMPASS mutants are not large (*Figure 1C*), so we pooled samples in the initial analysis to increase statistical power. Given that all the tested COMPASS mutants reduced lifespan and may therefore have similar effects, we pooled together the replicate datasets for *swd1Δ*, *set1*$^{H1017L}$ and *spp1Δ* for comparison to wild type (giving 7 – 9 COMPASS mutant replicates per time point, compared to five wild-type replicates), and used DEseq2 to calculate significantly differentially expressed genes at each time point (summarised in Table S1). 107 genes were significantly different between wild type and COMPASS mutants at log phase, with more being differentially expressed at 24 and 48 hr (389 and 488), but few at 7.5 hr (39). Interestingly, log phase cells contain a subset of genes that are

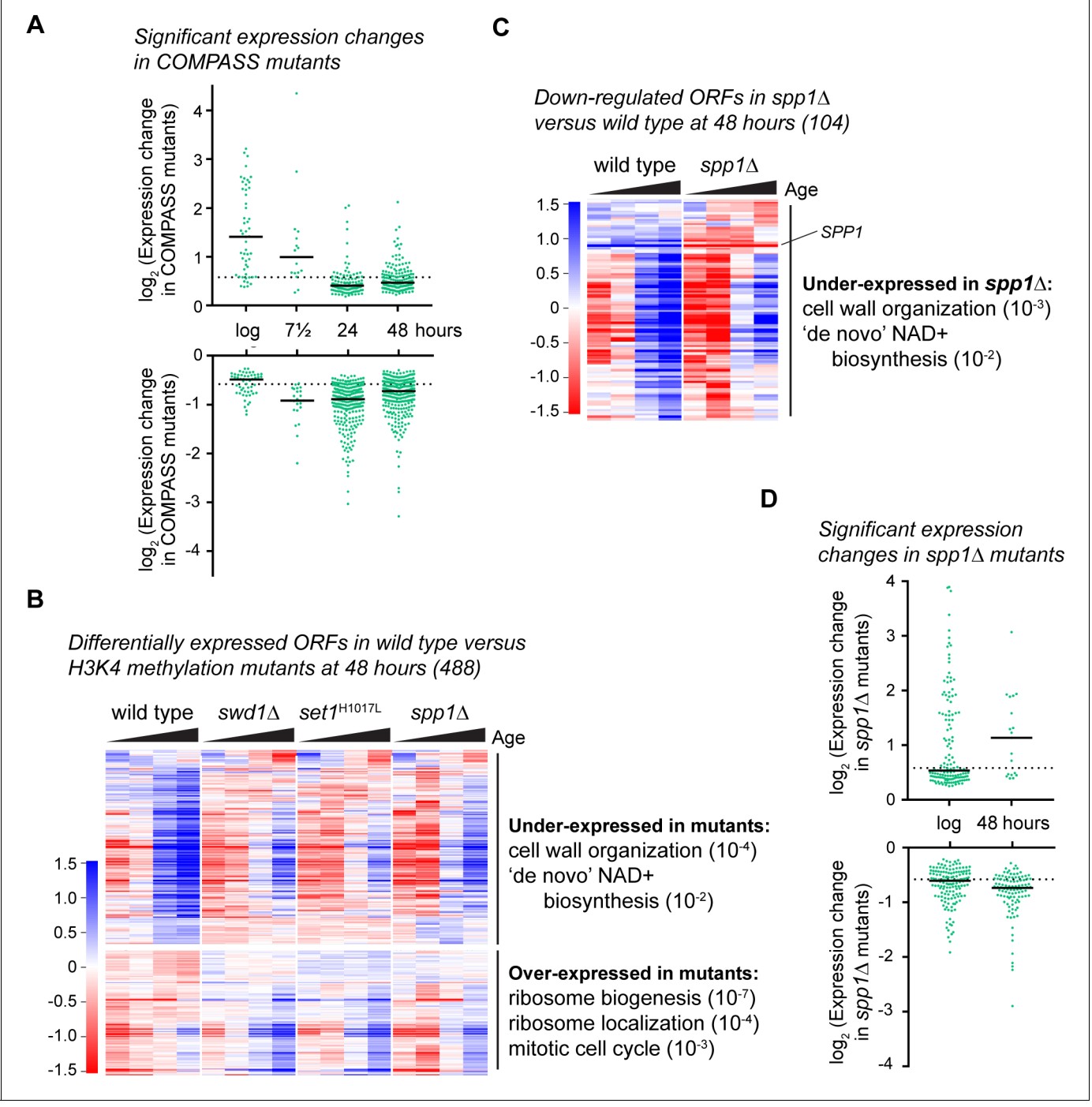

**Figure 2.** COMPASS is required for normal gene expression in ageing cells. (**A**) Plot of $log_2$-transformed read-count differences for significantly differentially expressed genes between wild type and COMPASS mutants at each ageing time point. Assessed using DEseq2 p<0.01, wild-type n = 5 at each time point, COMPASS mutants n = 8 at log, n = 9 at 7.5 hr, n = 7 at 24 hr and n = 8 at 48 hr. Horizontal bars show median, dotted line indicates 1.5-fold change. (**B**) Hierarchical clustering analysis of $log_2$-transformed protein-coding mRNA levels across age for the 488 genes significantly differentially expressed between wild-type and pooled COMPASS mutants at 48 hr, based on DEseq2 analysis in A. Separate time-courses are given for wild-type and each COMPASS mutant to show similar behaviour, GO analysis as for *Figure 1—figure supplement 2A*. (**C**) Hierarchical clustering analysis of $log_2$-transformed protein-coding mRNA levels across age for the 104 genes significantly down-regulated in *spp1Δ* cells compared to wild-type at 48 hr, assessed using DESeq2 p<0.05 n = 5 per strain. GO analysis as *Figure 1—figure supplement 2A*. (**D**) Distribution of significantly differentially expressed genes between wild type and *spp1Δ* at log and 48 hr, analysis as in A, n = 5 for each set.

DOI: https://doi.org/10.7554/eLife.34081.006

*Figure 2 continued on next page*

*Figure 2 continued*

The following figure supplement is available for figure 2:

**Figure supplement 1.** Supplement to COMPASS is required for gene expression in ageing cells.
DOI: https://doi.org/10.7554/eLife.34081.007

substantially over-expressed in COMPASS mutants (on average 3-fold), but this difference disappears with age (*Figure 2A*, top). In contrast, significantly under-expressed genes at log phase are on average only <1.5 fold lower than in wild type, but with age many more genes become under-expressed in COMPASS mutants and to a greater degree (2-fold on average, but many are 3 – 4-fold reduced) (*Figure 2A*, bottom), suggesting that the role of COMPASS in gene expression is age dependent.

This data are complicated by the fact that almost all of these genes are induced with age in all genotypes (*Figure 2—figure supplement 1A*), but the extent of induction varies between genotypes. Therefore, genes that are over-expressed at log phase in COMPASS mutants (specifically *swd1Δ* and *set1*^H1017L) are also over-expressed in aged COMPASS mutant samples relative to an age-matched wild type but the difference between wild type and mutant decreases with age. In contrast genes that are under-expressed in COMPASS mutants at log phase are under-expressed to a similar or greater extent throughout ageing compared to age-matched controls (*Figure 2—figure supplement 1A*). These log phase data are in accord with previous results (*Margaritis et al., 2012*), but it is clear that the importance of COMPASS for gene expression changes with increasing age; in general, the repressive function of COMPASS has less impact and a positive function in gene expression is unmasked.

Starting with the 48 hr time point, we segregated under- and over-expressed genes and performed GO analyses (*Figure 2B* and Table S1). Over-expressed genes (191) were primarily enriched for ribosome-related terms as expected (*Weiner et al., 2012*), but under-expressed genes (297) were enriched for *cell wall organisation* and *de novo NAD+ biosynthesis*, both of which are known to be important for maintenance of replicative lifespan (*Bonkowski and Sinclair, 2016*; *Molon et al., 2018*). Although these genes were identified using a pooled dataset from three COMPASS mutants, all the COMPASS mutations had a similar effect on expression (*Figure 2—figure supplement 1B*). Furthermore, an equivalent analysis comparing wild type to *swd1Δ* identified 240 under-expressed genes at 48 hr of which 60% overlapped with those identified in the pooled COMPASS mutant dataset (*Figure 2—figure supplement 1C* and Table S1), and GO analysis of these 240 genes also indicated effects on cell wall biosynthesis and nicotinamide metabolism. Notably, in comparisons of wild-type to pooled COMPASS mutants or *swd1Δ*, the majority of genes affected by loss of COMPASS failed to be properly upregulated with age; this behaviour has not been observed in other transitions – Margaritis *et al.* examined the massive transcriptional reprogramming that accompanies the transition from stationary phase to log phase growth and found 220 genes mis-regulated of which only 24 (10%) were under-expressed in COMPASS mutants (*Margaritis et al., 2012*), compared to 297 (61%) of significantly altered genes that we observe in aged cells.

Differential expression in the pooled dataset could be attributed to mono-, di- or trimethylation of H3K4. To discover effects stemming purely from trimethylation, we sequenced three additional *spp1Δ* samples at log phase and after 48 hr ageing, giving five replicates in total. We employed a more relaxed significance threshold (p<0.05) for differential expression analysis to reflect the fact that *spp1Δ* is not a complete loss-of-function mutant as some residual H3K4me3 is still present (*Figure 1—figure supplement 1B*) and is therefore expected to cause a weaker phenotype; this yielded 104 significantly under-expressed genes in aged *spp1Δ* cells compared to wild type along with 19 over-expressed genes (*Figure 2C*). Despite the relaxed significance threshold, the under-expression phenotype remained highly consistent across replicates (*Figure 2—figure supplement 1D*). GO analysis of the under-expressed genes again revealed significant enrichments for *cell wall biogenesis* and *de novo NAD+ biosynthesis* just as in the combined COMPASS mutant set, showing that proper expression of these genes is dependent on H3K4me3 (*Figure 2C*). Surprisingly, we also detected almost 300 genes differentially expressed between wild-type and *spp1Δ* in log-phase cells although the bulk of these showed very small differences (<1.5 fold), in contrast to the 48 hr samples for which

most differentially expressed genes were under-expressed and to a larger extent (>1.5 fold) (*Figure 2D*).

The H3K4 trimethylation activity of COMPASS depends on Spp1 in vivo but not in vitro raising the possibility that loss of Spp1 may have different effects in aged cells, either not suppressing H3K4 trimethylation or impairing H3K4 dimethylation. To confirm that COMPASS activity is globally similar in young and old cells we probed western blots of log phase and 24 hour-aged wild-type and *spp1Δ* cells for H3K4me2 and H3K4me3. This confirmed that H3K4me2 levels were similar in both strains in young and old cells, and that Spp1 is also required for H3K4me3 in old cells (*Figure 2—figure supplement 1E*).

Overall, although COMPASS mutants are considered to have little impact on gene expression in log phase, we detect many genes (~4.5%) that require COMPASS to attain normal expression over the period of the yeast replicative lifespan.

## Importance of H3K4me3 in NAD+ biosynthesis gene expression

The failure of many genes to attain full expression during ageing in H3K4 methylation mutants indicates a role for this mark in facilitating gene expression. However, the complexity of the ageing process and the short lifespan of COMPASS mutants may render such effects indirect. To validate this association we focused on the *BNA* genes, which are regulated in a fairly simple manner by promoter binding of the repressive NAD+-dependent histone deacetylase Hst1 (*Figure 3A*) (*Bedalov et al., 2003*). *BNA1*, *BNA2* and *BNA4-7* encode the enzymes required for biosynthesis of the NAD+-precursor NaMN from tryptophan (*Figure 3A*), and differential expression of these genes underlies the GO enrichment for *de novo biosynthesis of NAD+* in ageing COMPASS mutants described in the previous section.

We first assessed the expression of these genes individually across age in the RNAseq dataset. *BNA1*, *BNA2*, *BNA4*, *BNA5* and *BNA6* are all induced with age and this induction is impaired in *spp1Δ*, with *BNA7* forming an outlier that is unchanged by age or *SPP1* deletion (*Figure 3B* compare wild type to *spp1Δ*). The effect of *swd1Δ* and *set1*[H1017L] is more complex: *BNA1*, *BNA5* and *BNA6* also show a defect in aged-linked induction in these mutants; *BNA2* and *BNA4* induce less but from a higher basal level; *BNA7* shows reduced basal expression. These data reveal that loss of H3K4me3 has a different effect on the *BNA* genes to loss of all H3K4 methylation.

Usefully, the direct negative regulation of the *BNA* genes by Hst1 allowed us to quantify induction in non-ageing log-phase cells through treatment with the Hst1 inhibitor nicotinamide (*Figure 3A*, Hst1 is subject to product inhibition by nicotinamide). As ageing was not required, we performed these experiments in standard BY4741 haploid cells without the substantial genetic modifications required for the MEP system. Five replicate sets of wild-type, *spp1Δ* and *swd1Δ* cells were grown in YPD, treated for 6 hr ± 1 mM nicotinamide, RNA was extracted and expression of *BNA1* and *BNA4-7* determined in treated and untreated cells relative to ribosomal RNA by northern blot.

The induced gene expression level of *BNA1*, *BNA4* and *BNA5* was significantly reduced in *spp1Δ* cells, with *BNA6* showing a similar trend (*Figure 3C,D*, *Figure 3—figure supplement 1A,B* and *Figure 3—figure supplement 2A*). Furthermore *BNA1*, *BNA5* and *BNA6* were significantly under-expressed even in the absence of nicotinamide treatment. Therefore, while the reduced expression of *BNA4* in nicotinamide-treated *spp1Δ* cells can be attributed solely to decreased induction, *BNA1* and *BNA5* mRNA levels are lower in nicotinamide-treated *spp1Δ* cells due to a combination of weaker induction and reduced basal expression. In contrast, *BNA7* was not induced by nicotinamide and was unaffected in *spp1Δ* cells (*Figure 3—figure supplement 1C* and *Figure 3—figure supplement 2A,C*). These results matched the RNAseq data for log and 48hr-aged cells remarkably well (*Figure 3C,D* and *Figure 3—figure supplement 1A,B,C* compare left and right panels), showing that the impact of H3K4me3 is not restricted to ageing cells and therefore that H3K4me3 is required to maintain both basal and induced expression levels of the *BNA* genes.

The *BNA* genes are induced to a lesser extent in *swd1Δ* mutants compared to wild type but the effect is less clear-cut as basal expression is more variable, ranging from much higher than wild type in *BNA4* to lower than wild type in *BNA6* (*Figure 3C,D*, *Figure 3—figure supplement 1A,B* and *Figure 3—figure supplement 2A*). The basal expression level of *BNA7* is also reduced in *swd1Δ* mutants and is not induced back to wild-type levels by nicotinamide (*Figure 3—figure supplement 1C* and *Figure 3—figure supplement 2A,C*). Again, these observations closely paralleled the ageing RNAseq data at 48 hr, and combined with the results from *spp1Δ* cells show that different H3K4

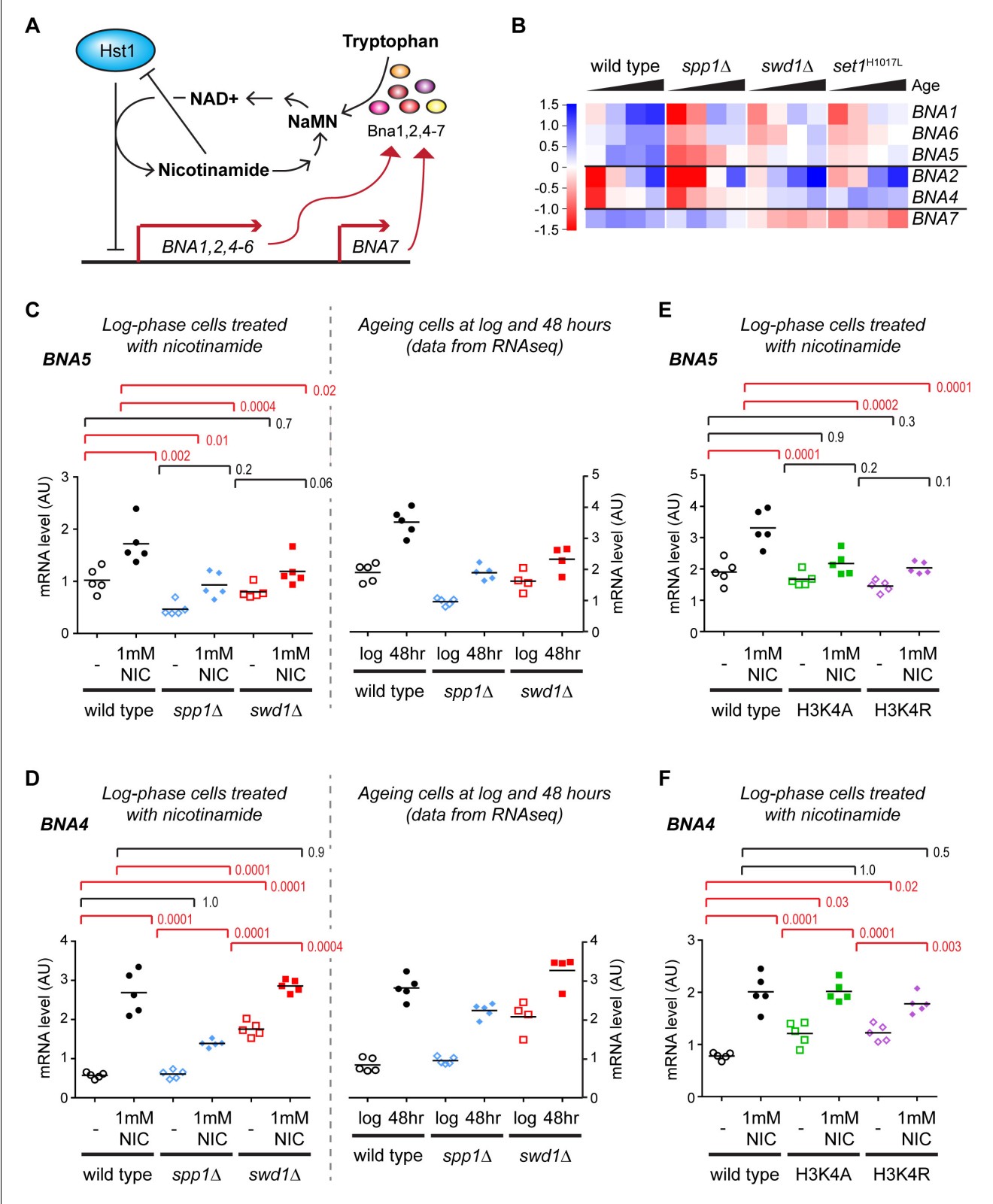

**Figure 3.** Validation of gene expression differences in COMPASS mutants. (**A**) Schematic representation of the regulatory feedback system controlling expression of the *BNA* genes which encode enzymes in the pathway for NAD+ biosynthesis from tryptophan via nicotinic acid mononucleotide (NaMN). Essentially, Hst1 uses NAD+ as a cofactor to repress the *BNA* genes, so that when NAD+ is low the expression of the *BNA* genes rises to increase NAD+ biosynthesis. NAD+ is converted to nicotinamide by Hst1, which is a product inhibitor and limits the repressive activity. Exogenous

*Figure 3 continued*

nicotinamide is efficiently taken up by cells, allowing Hst1 inhibition in culture. NAD+ can also be synthesised from nicotinamide riboside, but this does not involve the BNA genes. (B) Hierarchical clustering analysis of log$_2$-transformed protein-coding mRNA change for genes in the tryptophan to NaMN biosynthesis pathway in wild type and COMPASS mutants across ageing. (C) Northern blot quantification of *BNA5* mRNA level relative to ribosomal RNA in log phase BY4741 haploid cells grown in YPD and treated for 6 hr ± 1 mM nicotinamide (left panel) compared to mRNA abundance measurements for *BNA5* derived from sequencing data for log and 48 hr aged cells (right panel). p-values were calculated by one-way ANOVA, significant differences are highlighted in red, n = 5 for each category. (D) Analysis of *BNA4* expression as in C. (E) Analysis of *BNA5* expression in histone point mutants ± nicotinamide, performed as in C. (F) Analysis of *BNA4* expression in histone point mutants ± nicotinamide, performed as in C.

DOI: https://doi.org/10.7554/eLife.34081.008

The following figure supplements are available for figure 3:

**Figure supplement 1.** Supplement to validation of gene expression differences in COMPASS mutants.

DOI: https://doi.org/10.7554/eLife.34081.009

**Figure supplement 2.** Example northern blotting data.

DOI: https://doi.org/10.7554/eLife.34081.010

---

methylation states have different effects on gene expression for the *BNA* genes. We propose a model for this behaviour in the Discussion.

COMPASS has at least one non-histone substrate (*Zhang et al., 2005*), so to ensure that the effects on NAD+ biosynthesis genes result from H3K4 methylation we performed further experiments in histone mutants H3K4A and H3K4R. As before we used nicotinamide to induce expression of the *BNA* genes in haploid cells growing at log phase in YPD, and observed that the H3K4 mutants largely phenocopied the COMPASS mutants; H3K4 mutant phenotypes for *BNA4-7* are very similar to *swd1Δ*, while the phenotype for *BNA1* lies in-between that of *swd1Δ* and *spp1Δ* (*Figure 3E,F*, *Figure 3—figure supplement 1D–F* and *Figure 3—figure supplement 2B*). This is consistent with COMPASS acting through H3K4 methylation to alter the expression of the *BNA* genes.

Together, these data show that COMPASS mutation affects the expression of NAD+ biosynthesis genes directly via H3K4 methylation, rather than through indirect age-induced changes in metabolism. They further provide strong evidence for a critical role of H3K4me3 in maintaining the normal expression of yeast genes. Although the 2-fold under-expression observed here in *spp1Δ* mutants is not a huge effect, *BNA* gene mRNA levels are tightly auto-regulated in response to NAD+ levels so this is likely to limit the production of NAD+ for the lifetime of the organism.

## Age-linked changes in H3K4me3 distribution

Ageing in yeast is accompanied by widespread gene expression changes and a shift in chromatin structure. To understand how these differences affect individual promoters requires the genome-wide distribution of H3K4me3 to be determined, however mapping histone modifications in ageing yeast cells has proved extremely challenging as standard yeast chromatin immunoprecipitation (ChIP) protocols require 100-1000x more cells than are available from routine ageing cell purifications (explaining the very limited genome-wide ageing ChIP data available for yeast [*Hu et al., 2014*; *Sen et al., 2015*]). To facilitate the analysis of ageing chromatin, we optimised a low-cell number yeast ChIP protocol that yields ChIPseq libraries from the 10$^6$ cells available through our standard ageing cell preparation methods, including a spike-in of *Drosophila* chromatin prior to cell lysis that facilitates absolute quantification of signals. Using this protocol, we obtained ChIPseq datasets for H3 and H3K4me3 in log phase cells and cells aged for 24 and 48 hr in duplicate, along with matched input DNA. Input samples confirmed known age-linked genetic changes including amplification of ribosomal DNA (rDNA) and telomeres (*Sinclair and Guarente, 1997*), amplification of Chr. XII distal to the rDNA and amplification of Ty elements (predicted given increase in Ty element RNAs)(*Hu et al., 2014*) (*Figure 4—figure supplement 1A*), and all of these regions were therefore excluded from the initial chromatin analysis (Table S2). However, detection of these differences confirmed that ChIP samples derived from highly aged cells rather than contaminating daughters. We also restricted analysis of H3K4me3 peaks to genes for which the region TSS to TSS+ 500 bp could be unambiguously assigned by excluding short sense and antisense ORFs that overlap promoter regions. Overall, these limits define a set of 4806 genes which reflect well the spectrum of expression observed across all genes but for which copy number abnormalities and promoter ambiguities were minimised (compare *Figure 4—figure supplement 1B* to *Figure 1—figure supplement 2B*).

With age, the signal from promoter associated H3K4me3 peaks increased and the spread of H3K4me3 signals was compressed (*Figure 4A*). Particularly notable is the progressive loss of repressed promoters with no detectable H3K4me3 peak (*Figure 4A* grey arrows). These observations are consistent with H3K4me3 peaks at inactive or low activity promoters becoming more prominent as would be expected from the genome-wide induction of low-expressed genes, and indeed we observe a strong positive correlation between age-linked gene induction and change in promoter H3K4me3 signal (*Figure 4B*). By 48 hr, the promoter H3K4me3 peak also became less defined, due to increasing H3K4me3 in gene bodies and intergenic regions (*Figure 4C*). This behaviour is somewhat variable across genes: for the 300 least induced genes, which are highly expressed throughout life, we observe on average a reduction of the H3K4me3 peak between 24 and 48 hr (*Figure 4—figure supplement 1C*), but in contrast the 300 most induced genes, which have very low log phase expression, show a dramatic age-linked increase in H3K4me3 in the region round the TSS as well as the appearance of a small H3K4me3 peak (*Figure 4—figure supplement 1D*). We suspected that the general increase in H3K4me3 in non-promoter regions would reflect a widespread activation of cryptic promoters and spurious transcriptional initiation, leading to extensive pervasive transcription. To assess this, we quantified antisense poly(A)+RNA for annotated genes and found that antisense transcripts increase substantially with age and to a similar extent for all genes, despite clear differences in age-linked expression change for the sense mRNA (compare *Figure 4D* and *Figure 4—figure supplement 1E*). This upregulation of antisense transcription independent of sense transcript expression is indicative of a genome-wide increase in pervasive transcription. Overall, age-linked changes in promoter-associated H3K4me3 are consistent with genome-wide gene induction, but the tight promoter association of H3K4me3 is compromised in old cells and H3K4me3 accumulates in non-promoter regions in accord with an observed increase in pervasive transcription.

To better understand global changes linked to ageing, we assessed the density of H3K4me3 and H3 using normalised ChIP signals in 1 kb windows across the whole genome (excluding mitochondrial DNA) and correcting for differences in copy number based on the input DNA. The rDNA was analysed separately as this region is massively amplified during ageing; we observe a 7-fold rDNA copy number increase from log to 48 hr based on ChIP input material (*Figure 4—figure supplement 2A*), consistent with known accumulation of extrachromosomal rDNA circles (ERCs) and other extrachromosomal rDNA species (*Pal et al., 2018*; *Sinclair and Guarente, 1997*). Non-rDNA regions gain H3K4me3 between log and 48 hr of age as noted above, while rDNA H3K4me3 shows a striking rise from very low levels at log phase to the same level as non-rDNA regions by 48 hr consistent with a complete loss of rDNA silencing (*Figure 4E*) (*Li et al., 2017*; *Pal et al., 2018*). Changes in H3 are less dramatic, with a mild reduction in H3 in non-rDNA regions and little difference in the rDNA (*Figure 4—figure supplement 2B*). However, the massive increase in rDNA copy number results in unexpected changes in global distributions of both H3K4me3 and H3: the combination of loss of rDNA silencing with increased copy number leads to the total amount of H3K4me3 in the genome increasing 2-fold by 48 hr, with the proportion of H3K4me3 in rDNA rising from 2.5% to 40% (*Figure 4F*). This is accompanied by a small (~1.3 fold) increase in H3, again driven by increases in rDNA copy number (*Figure 4—figure supplement 2C*). Taken together, these results implied that the amount of H3K4me3 in the cell should rise with age relative to H3, an effect that we confirmed independently by western blot (*Figure 4G*).

These data show that age-associated genome-wide gene induction is accompanied by rising promoter H3K4me3, and that H3K4me3 accumulates both in non-promoter regions and in the rDNA as cells age, associated with loss of rDNA heterochromatin and increased genome-wide pervasive transcription.

## Discussion

The predicted role of H3K4me3 in facilitating transcription has proved challenging to validate. Here, we have shown that a large subset of yeast genes depend on H3K4me3 to attain normal expression levels. This effect becomes prevalent during the ageing process in which huge numbers of genes are induced coincident with a decline in the repressive capacity of H3K4 methylation. Our results provide a clear demonstration that H3K4me3 is indeed an epigenetic mark that facilitates gene expression, and one that plays a critical role in maintaining cell viability across the life-course.

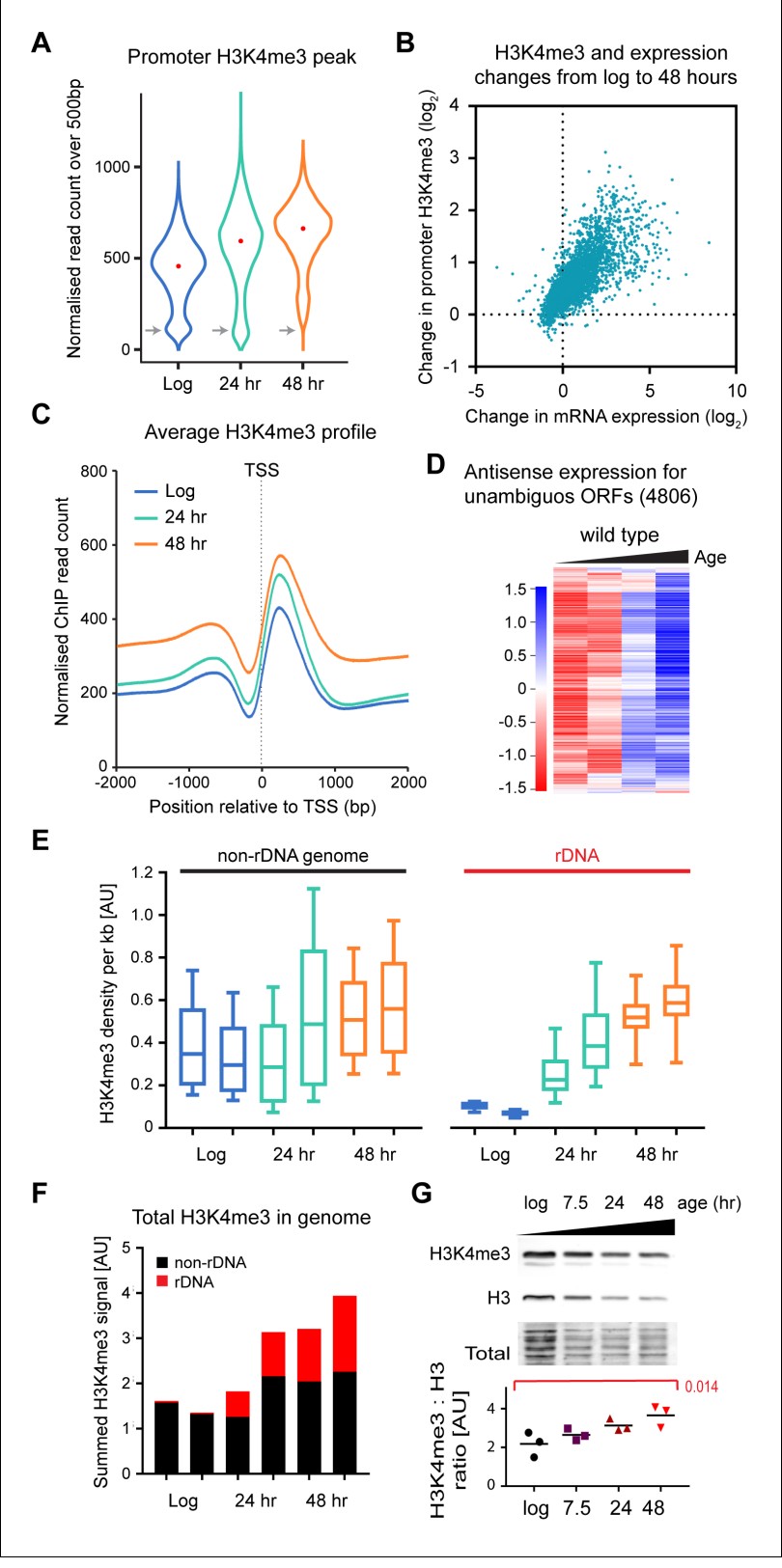

**Figure 4.** Age-linked dynamics of H3K4me3. (**A**) Violin plot of read counts for region from transcriptional start site (TSS) to TSS+ 500 bp for all genes lying outside excluded regions defined in *Figure 4—figure supplement 1A* and with unambiguous promoter regions (4806 genes). Read counts were normalised using *Drosophila* chromatin spike-ins, data are an average of two biological replicates at each time point. Grey arrows highlight category of

*Figure 4 continued on next page*

*Figure 4 continued*

genes with very low H3K4me3 that disappears with age. (**B**) Plot of the change in H3K4me3 (TSS to TSS + 500 bp) from log to 48 hr for each gene in (**A**) versus change in mRNA level from log to 48 hr. Data are an average of two biological replicates for H3K4me3 and five biological replicates for mRNA. (**C**) Average H3K4me3 signal across region of TSS ± 2 kb for all genes in (**A**), profiles are normalised to *Drosophila* chromatin spike-ins. (**D**) Hierarchical clustering analysis of log$_2$-transformed antisense poly(A)+RNA levels across age for the 4806 unambiguous genes defined in (**A**). (**E**) Average H3K4me3 density in 1 kb windows across rDNA and non-rDNA regions of the genome. Within each window, spike-in normalised ChIP read count was divided by spike-in normalised input read count to account for differences in copy number. Both replicates are shown for each time-point. (**F**) Summed normalised read counts across the genome for H3K4me3, sub-divided into rDNA and non-rDNA sequences. (**G**) Western blot analysis of histone H3 and H3K4me3 in wild-type cells aged for 0, 7.5, 24 or 48 hr, performed using a 2-colour system to simultaneously detect both antigens. Ratios of H3K4me3 to H3 were compared by one-way ANOVA, n = 3, specific comparison and p-value is shown in red.

DOI: https://doi.org/10.7554/eLife.34081.011

The following figure supplements are available for figure 4:

**Figure supplement 1.** Dynamics of H3K4me3 and expression in subsets of genes.

DOI: https://doi.org/10.7554/eLife.34081.012

**Figure supplement 2.** Age linked changes in genome size and H3 distribution

DOI: https://doi.org/10.7554/eLife.34081.013

## Gene expression outcomes of H3K4 methylation

H3K4 methylation has a remarkably complex effect on gene expression. Transcriptomic approaches have detected repressive functions (*Guillemette et al., 2011*; *Jaiswal et al., 2017*; *Lenstra et al., 2011*; *Margaritis et al., 2012*; *Venkatasubrahmanyam et al., 2007*), although in most cases this can probably be attributed to H3K4me2 with the possible exception of ribosomal protein genes (*Weiner et al., 2012*). Evidence that H3K4me3 can promote gene expression in yeast is rarer; this was reported in early studies but has not been effectively reproduced (*Miller et al., 2001*; *Santos-Rosa et al., 2002*), see (*Margaritis et al., 2012*) for a discussion of technical reasons for this. Setting aside technical issues, if H3K4 methylation has both repressive and inducing activities, the outcome of COMPASS mutations on individual genes would be expected to vary by gene, by genetic background and by experimental system, making broad conclusions rather treacherous.

Antagonism between activating and repressive activities would help explain the conflicting data in the literature, and close examination of the data in *Figure 3* provides support for this: loss of Spp1 reduces expression for most of the *BNA* genes, whereas loss of Swd1 has a milder and more variable effect that likely results from the summed outcomes of repressive H3K4me2 and activating H3K4me3 (see summary of H3K4me3 and H3K4me2 effects on the *BNA* genes in *Figure 5*). However for other genes, the balance and direction of these effects is clearly different, for example at *BNA7* H3K4me2 appears to be activating while H3K4me3 has no effect. Therefore, the impact of H3K4 methylation cannot be generalised but must be considered on a gene-by-gene basis taking into account both repressive and activating effects.

How might H3K4me3 promote gene expression? Multiple H3K4me3-binding proteins are present in yeast, which in turn are members of histone modifying and repositioning complexes. Binding of Yng1, Yng2 or Sgf29 would recruit NuA3, NuA4 or SAGA histone acetyl transferase complexes respectively, promoting local histone acetylation that should enhance transcriptional activation (*Agalioti et al., 2002*; *Bian et al., 2011*; *Pokholok et al., 2005*; *Steunou et al., 2016*; *Taverna et al., 2006*). Loss of recruitment of NuA3 and NuA4 to H3K4me3 has remarkably little effect on gene expression under normal conditions (*Choy et al., 2001*; *Taverna et al., 2006*), but may well impact a specific subset of genes when induced during environmental change or during dramatic chromatin remodelling of the sort that occurs during ageing; indeed delays in induction of individual genes have been reported in mutants lacking H3K4me3-linked acetyltransferase recruitment (*Bian et al., 2011*; *Morillon et al., 2005*).

## The impact of H3K4 methylation on ageing cells

Our experiments very clearly show that H3K4 methylation and in particular H3K4me3 is critical for the maintenance of a normal lifespan. We observed 43% of yeast genes to be differentially

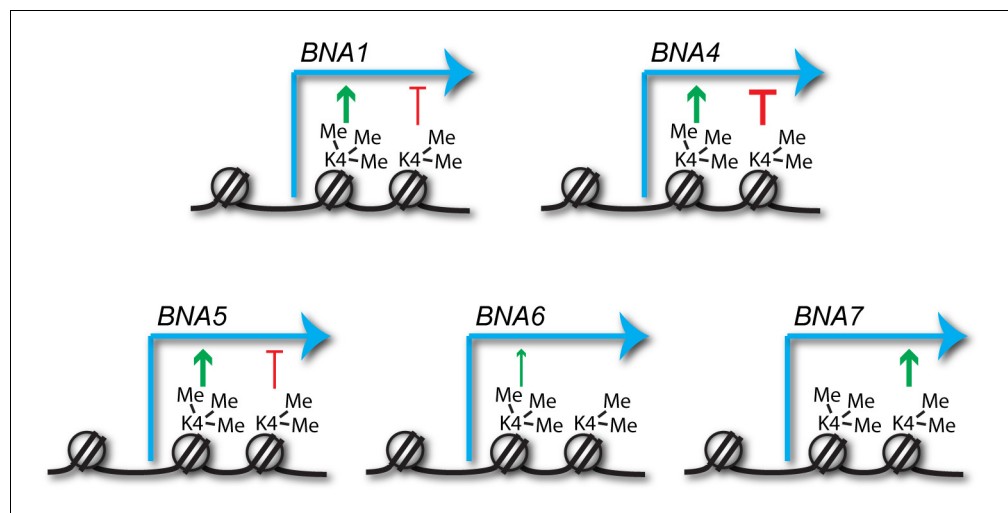

**Figure 5.** Predicted impacts of H3K4me2 and H3K4me3 on *BNA* gene expression. Schematic of the different impacts of H3K4me2 and H3K4me3 on expression of the *BNA* genes analysed in *Figure 3* and *Figure 3—figure supplement 1*. Depiction of these modifications on the +2 and+1 nucleosomes respectively is for illustrative purposes only and simply indicates that H3K4me2 normally spreads further into the gene body than H3K4me3.
DOI: https://doi.org/10.7554/eLife.34081.014

expressed during ageing, of which 17% show significantly different expression levels in COMPASS mutants after 48 hr. This means that as cells age, loss of H3K4 methylation has a dramatic effect on maintenance of normal gene expression patterns. We observed a progressive loss of the repressive capacity of H3K4 methylation as cells age, however this is probably not the cause of the lifespan defect, as the repressive activity is not impacted by *spp1Δ* whereas lifespan is similar in *swd1Δ* and *spp1Δ* cells. Rather, the most prominent gene expression effect was a failure to fully induce hundreds of genes in COMPASS mutants including *spp1Δ*. We validated this role of COMPASS for expression of NAD+ biosynthesis genes, but these constitute only a handful of the significantly mis-expressed genes and are unlikely to explain the shortened lifespan of COMPASS mutants alone. In reality, given the widespread age-linked gene mis-regulation in COMPASS mutants, it is not surprising that the lifespan of these cells is profoundly reduced.

ChIP experiments show that H3K4me3 is progressively gained at euchromatic promoters in accord with genome-wide gene induction, despite a mild reduction in H3 density. Spike-in normalisation was critical for obtaining accurate measurements of H3K4me3 as normalisation to internal standards erroneously shows loss of H3K4me3 with age, emphasising the importance of normalisation controls in ageing studies (*Chen et al., 2015*). The increase in H3K4me3 across euchromatic promoters is however dwarfed by the massive accumulation of H3K4me3 in rDNA, resulting in >2 fold increase in H3K4me3 across the genome. As noted above, H3K4me3 is involved in recruitment of pro-expression factors, but repressors such as the Set3 and Rpd3 complexes are recruited to H3K4me2 (*Kim and Buratowski, 2009*; *Pinskaya et al., 2009*). It is largely this age-linked loss of repressive activity that allowed us to detect pro-expression activities of H3K4me3, and it is tempting to speculate that accumulation of H3K4me2 in amplified rDNA (which is strongly predicted based on our data) would titrate repressive factors. Similarly, although the density of H3 in euchromatin decreases in aged samples in our experiments as previously reported (*Hu et al., 2014*), we find that this is more than offset by the increase in H3 associated with rDNA. It is not hard to imagine that cells struggle to maintain histone supply in the face of the almost doubling of genome size caused by profligate rDNA amplification; although different groups have reported either decreased or unchanging histone supply with age (*Feser et al., 2010*; *Janssens et al., 2015*), there is no indication that supply increases to match demand caused by rDNA amplification. Although ERC formation has been widely suggested to drive ageing in yeast, there is little mechanistic data regarding the potential pathological activity of ERCs. It seems very likely to us that one result of rDNA

amplification through ERC accumulation is to titrate histones and histone binding factors, and this will provide an important focus for future studies into age-linked gene expression change.

## Materials and methods

### Strains and media

Yeast strains were constructed by standard methods and are listed in Table S3, oligonucleotide sequences are given in Table S4 and plasmids given in Table S5. To construct $set1^{H1017L}$, H3$^{K4A}$ and H3$^{K4R}$ we used the break-mediated *Delitto Perfetto* system (Storici and Resnick, 2006): after integration of the CORE cassette in *SET1* or *HHT2*, *SceI* expression was induced using 2% galactose and oligonucleotides OCC339/340, OCC379/380 and OCC381/382 respectively transformed in followed by selection for fluoroorotic acid and against G418. For H3 mutants, the *HHT1-HHF1* locus was then deleted. All cells were grown in YPD media (2% peptone, 1% yeast extract, 2% glucose) at 30°C with shaking at 200 rpm. Media components were purchased from Formedium and media was sterilised by filtration. For MEP experiments, cells were inoculated in 4 ml YPD and grown for 6 – 8 hr then diluted in 25 ml YPD and grown for 16 – 18 hr to $0.2 – 0.6 \times 10^7$ cells/ml. If required, cells were labelled (see below), then cells were inoculated in YPD at $2 \times 10^4$ cells/ml containing 5 μg/ml ampicillin (Melford A0104) and 1 μM β-estradiol (Sigma E2758). For viability assay time points, 10 μl of culture was diluted with 40 μl water and spread on a YPD plate.

### MEP cell labelling and purification

For biotin labelling, $0.25 \times 10^7$ cells per sample were harvested by centrifugation (15 s at 13,000 g), washed twice with 125 μl PBS and re-suspended in 125 μl of PBS containing ~3 mg/ml Biotin-NHS (Sigma B1022 or Pierce 10538723). Cells were incubated for 30 min on a wheel at room temperature, washed once with 125 μl PBS and re-suspended in YPD. For 7.5 hr time points, cells were inoculated in 15 ml YPD +5 μg/ml ampicillin and grown for 2 hr before addition of 1 μM β-estradiol (Sigma E2758). For 24 and 48 hr time points, cells were directly inoculated in 125 ml YPD containing 5 μg/ml ampicillin and 1 mM β-estradiol. For 24/48 hr time points, no contamination of labelled non-aged cells was observed by bud scar staining as these cells presumably lyse, and this strategy helped keep the culture density low (cultures did not exceed $0.75 \times 10^7$ cells/ml in 48 hr). Cells were harvested by centrifugation for 1 min, 4600 rpm in 15/50 ml tubes and immediately fixed by resuspension in 70% ethanol and storage at −80°C. Alternatively, formaldehyde (Thermo 11586711) was added to 1% and cells incubated at room temperature for 15 min before quenching with 150 mM glycine, harvested by centrifugation as above and washed three times with PBS before freezing on N$_2$ and storage at −80°.

Rapid purification for RNA or protein: Percoll gradients (1 for 7.5 hr or 2 for 24/48 hr time points) were formed by vortexing 1 ml Percoll (Sigma P1644) with 110 μl 10x PBS in 2 ml tubes and centrifuging 15 min at 16,000 g, 4°C. Ethanol-fixed cells were defrosted and washed 1x with 1 vol of cold PBSE (PBS + 2 mM EDTA) before resuspension in ~250 μl cold PBSE per gradient and layering on top of the pre-formed gradients. Gradients were centrifuged for 4 min at 2,000 g, then the upper phase and brown layer of cell debris removed and discarded. 1 ml PBSE was added, mixed by inversion and centrifuged 1 min at 2,000 g to pellet the cells, which were then re-suspended in 1 ml PBSE per time point (re-uniting samples split across two gradients). Cells were briefly sonicated in a chilled Bioruptor 30 s on low power, then 25 μl Streptavidin microbeads added (Miltenyi Biotech 1010007) and cells incubated for 5 min on a wheel at room temperature. Meanwhile, 1 LS column per sample (Miltenyi Biotech 1050236) was equilibrated with cold PBSE in 4°C room. Cells were loaded on columns and allowed to flow through under gravity, washed with 2 ml cold PBSE and eluted with 1 ml PBSE using plunger. Cells were re-loaded on the same columns after re-equilibration with ~500 μl PBSE, washed and re-eluted, and this process repeated for a total of three successive purifications. 50 μl cells were set aside for quality control, while the remainder were pelleted by centrifugation and processed directly for RNA or protein.

For quality control, the 50 μl cells were diluted to 300 μl final volume containing 0.01% triton X-100, 0.3 μl 1 mg/ml streptavidin 594 (Life Technologies S11227), 0.6 μl 1 mg/ml WGA-488 (Life Technologies W11261) and DAPI. Cells were stained for 15 min at room temperature in the dark,

washed once with PBS then re-suspended in 5 µl VectaShield (Vectorlabs H-1000). Purifications routinely yielded 80 – 90% streptavidin positive cells with appropriate bud-scar numbers.

Purification of formaldehyde-fixed cells for ChIP: Percoll gradients were formed as above. Cells were defrosted on ice, re-suspended in ~250 µl cold PBSE per gradient and layered on the preformed gradients. Gradients were centrifuged for 4 min at 2,000 g, then the upper phase and brown layer of cell debris removed and discarded. 1 ml PBSE was added, mixed by inversion and centrifuged 1 min at 2,000 g to pellet the cells, which were then re-suspended in 1 ml PBSE per time point (re-uniting samples split across two gradients). Cells were briefly sonicated in a chilled Bioruptor 30 s on low power, 25 µl Streptavidin magnetic beads were added (Miltenyi Biotech 1010007) and cells incubated for 5 min on a wheel at room temperature. Meanwhile, 1 LS column per sample (Miltenyi Biotech 1050236) was equilibrated with cold PBSE in 4°C room. Cells were loaded on columns and allowed to flow through under gravity, washed with 2 ml cold PBSE and eluted with 1 ml PBSE using plunger. Cells were re-loaded on the same columns after re-equilibration with ~500 µl PBSE, washed and re-eluted, and this process repeated for a total of three successive purifications. Final elution was to 1.5 ml tubes (Treff 1130153). 50 µl cells were set aside for quality control, while the remainder were adjusted to 0.01% Triton X-100 and pelleted by centrifugation and processed directly for ChIP.

## Low cell number RNA purification and library preparation

Cells were re-suspended in 50 µl Lysis/Binding Buffer (from mirVANA kit, Life Technologies AM1560), and 50 µl 0.5 µm zirconium beads (Thistle Scientific 11079105Z) added. Cells were lysed with 5 cycles of 30 s 6500 ms$^{-1}$ / 30 s ice in an MP Fastprep bead beater, then 250 µl Lysis buffer added followed by 15 µl miRNA Homogenate Additive and cells were briefly vortexed before incubating for 10 min on ice. 300 µl acid phenol: chloroform was added, vortexed and centrifuged 5 min at 13,000 g, room temperature before extraction of the upper phase. 400 µl room temperature ethanol and 1 µl glycogen (Sigma G1767) were added and mixture incubated for 1 hr at −30°C before centrifugation for 15 min at 13,000 g, 4°C. Pellet was washed with cold 70% ethanol and re-suspended in 10 µl nuclease free water. 1 µl RNA was glyoxylated and analysed on a BPTE mini-gel as described (*Sambrook and Russell, 2001*), and 0.2 µl quantified using a PicoGreen RNA kit (Life Technologies R11490).

500 ng RNA was used to prepare libraries using the NEBNext Ultra Directional mRNAseq kit (NEB E7420, E7490, E7335, E7500, E7710) as described with modifications: Reagent volumes for RT, second strand synthesis, tailing and ligation were reduced by 50%, while libraries were amplified for 12 cycles using 2 µl each primer per reaction before two rounds of AMPure bead purification as described in the kit at 0.9x ratio prior to quality control using a Bioanalyzer and quantification usinng KAPA Biosystems kit KK4824.

## Low cell number ChIP and library preparation

Cells were re-suspended in 50 µl cold CLB (50 mM HEPES pH 7.0, 140 mM NaCl, 1 mM EDTA, 1% Triton X100, 0.1% sodium deoxycholate, 0.1% SDS, cOmplete Protease Inhibitors [Roche 1836170]) with 3 ng *Drosophila melanogaster* spike-in chromatin (Active Motif 02317006) and 50 µl zirconium beads and lysed with 7 cycles of 30 s 6500 ms$^{-1}$ / 30 s ice in an MP Fastprep bead beater then transferred to a Covaris 520045 microTUBE and topped up to 130 µl with CLB. Sonication was performed on a Covaris 220: Duty Factor 5%, PIP 130 W, 900 s, 200 cycles per burst, 8.9°C. Samples were decanted to a 1.5 ml tube, centrifuged 5 min top speed at 4°C and the pellet discarded. Lysate was precleared by incubation with 15 µl of pre-equilibrated Protein-G magnetic beads (Life Technologies 10004D) 2 hr at 4°C on a wheel, then 30 µl set aside for total DNA extraction and 50 µl used per immunoprecipitation (IP). Antibodies were added at 1:50 dilution (rabbit anti-H3, CST 2650 or rabbit anti-H3K4me3, CST 9751) and to every IP 0.8 µg of spike-in antibody (Active Motif 61686) was added. IPs were incubated over night at 4°C on a wheel.

15 µl pre-equilibrated protein-G beads in 25 µl CLB were added to each IP and incubated 2 – 3 hr at 4°C on a wheel. Beads were then washed once each at room temperature for 5 min on a wheel with: CLB, CLB with 0.5 M NaCl, wash buffer (10 mM Tris pH 8.0, 0.25 M LiCl, 0.5% NP-40, 0.5% sodium deoxycholate, 1 mM EDTA), TE, then eluted overnight at 65°C with 40 µl elution buffer (50 mM Tris/HCl pH 8.0, 10 mM EDTA, 1% SDS). Input samples were treated with 1 µl 1 mg/ml RNase A

at 37°C for 1 hr then incubated overnight at 65°C. Input and IP samples were treated with 1 µl proteinase K for 2 hr at 55°C, then purified using ChIP DNA Clean and Concentrator with capped columns (Zymo D5205) and eluted in 10 µl

Libraries were synthesised using a NEBNext DNA Ultra II kit (NEB E7645) with modifications: Concentration of DNA was increased by reducing the volumes of the end repair and ligation steps 5-fold, with reagent quantities reduced in kind. After clean-up with 0.9x AMPure beads, PCR was performed in 50 µl volumes according to manufacturer's instructions. Libraries were purified with 0.9x AMPure beads and eluted in 30 µl, then a size selection was performed by incubating first with 15 µl AMPure beads, discarding the beads then purifying the remaining DNA from the supernatant with an additional 12 µl AMPure beads.

## Nicotinamide treatment, RNA purification and northern analysis

Cells were grown in 4 ml YPD for 6 hr, diluted and grown overnight in 25 ml YPD to $0.2 – 0.5 \times 10^7$ cells/ml. Cells were diluted to $0.06 \times 10^7$ cells/ml in 12.5 ml YPD and grown for 1.5 hr before addition of 12.5 ml YPD with or without 2 mM nicotinamide (Sigma N3376) and grown for 6 hr before harvesting $2 \times 10^7$ cells by centrifugation and freezing on $N_2$. RNA was extracted by the GTC phenol miniprep method, and 1.5 µl of 6 µl eluate run on 1% BPTE gels, imaged and blotted as described (*Cruz and Houseley, 2014*). Gels were probed for *BNA4* and *BNA6* then re-probed for *BNA1*, or for *BNA5* and *BNA7*, in UltraHyb (Life Technologies AM8670) using RNA probes listed in Table S6 as described (*Cruz and Houseley, 2014*).

## Protein purification and western blotting

Proteins were purified, separated and blotted as described with modifications (*Frenk et al., 2014*): aged MEP cells were re-suspended in Laemmli buffer containing 100 mM DTT and separated on 15% acrylamide gels, and Odyssey blocking buffer PBS (LI-COR 927 – 40000) was used for antibody hybridisations. Membranes were probed with antibodies listed in Table S7 at given dilutions before imaging on a LI-COR Odyssey 9120 imager. Total protein detection was performed by staining membranes with REVERT Total Protein Stain solution (LiCor 926 – 11010) after antibody detection.

## Image processing and data analysis

Northern blots images were quantified using ImageQuant v7.0 (GE), western blots using Aida v3.27.001 (Fuji). Images for publication were processed using ImageJ v1.50i, by cropping and applying minimal contrast enhancement. Statistical analysis of viability scores and northern/western data were performed using GraphPad Prism v7.03.

## Sequencing and bioinformatics

Libraries were sequenced by the Babraham Institute Sequencing Facility variously using HiSeq 2500 (RNAseq) or NextSeq 500 (ChIPseq) instruments. After adapter and quality trimming using Trim Galore (v0.4.2), RNAseq data were mapped to yeast genome R64-2-1 using HISAT v2.0.5 (*Kim et al., 2015*) by the Babraham Institute Bioinformatics Facility. Mapped data were imported into SeqMonk v1.39.0 (https://www.bioinformatics.babraham.ac.uk/projects/seqmonk/) and normalised based on the total number of reads mapping to the antisense strand of annotated open reading frames (opposite strand specific libraries), excluding the rDNA locus and the mtDNA. Exclusion of rDNA mapping reads from the normalisation is particularly important as in ageing cells a substantial fraction of reads derive from polyadenylated rDNA ncRNAs and can skew the distribution. DESeq2 analyses (*Love et al., 2014*) was performed within SeqMonk, comparing either five replicates of different time points in the wild-type dataset or comparing the five replicates of wild-type to pooled data from *swd1Δ*, *set1* and *spp1Δ* mutants at a given time point, using a p-value cut-off of 0.01 for analyses except comparisons of wild-type versus *spp1Δ* where a cut-off of 0.05 was applied. Hierarchical clustering analysis was performed using SeqMonk, and GO analysis of individual clusters performed using GOrilla (http://cbl-gorilla.cs.technion.ac.il/) (*Eden et al., 2007*; *Eden et al., 2009*). Quoted p-values for GO analysis are FDR-corrected according to the Benjamini and Hochberg method (q-values from the GOrilla output), for brevity only the order of magnitude rather than the full q-value is given (*Benjamini and Hochberg, 1995*). Full GO analyses are provided in *Supplementary file 2*. TSS data was obtained from *Pelechano et al. (2013)* – the strongest TSS in

GLU or GAL was used where TSS data were available, otherwise the start of the CDS was substituted.

After adapter and quality trimming using Trim Galore (v0.4.2), ChIPseq data were mapped to yeast genome R64-2-1 and *Drosophila* genome BDGP6 using Bowtie 2 (v2.3.2) by the Babraham Institute Bioinformatics Facility. Reads that mapped to the *Drosophila* genome were then remapped against the yeast genome and all cross-mapping reads discarded. Similarly cross-mapping reads from yeast to *Drosophila* were also discarded. For normalisation, in each dataset counts were performed of the total number of reads mapping to non-rDNA and non-mitochondrial regions of the yeast genome (Y) and the total number of uniquely mapping reads to the *Drosophila* genome (D). Normalisation factors were calculated using the formula $(1/D_{ChIP}) \times (D_{input}/Y_{input})$ and applied to yeast ChIP read counts. Mapped data were analysed using SeqMonk (v1.39.0, https://www.bioinformatics.babraham.ac.uk/projects/seqmonk/). Reads mapping to Excluded Regions (genomically amplified with age, see Table S2, *Figure 4—figure supplement 1A*) were excluded from the analysis. For histone density calculations (*Figure 4D–F*), the whole genome except for an expanded rDNA region (Chr XII 451000 – 491000) which contains multiple high copy regions and the mtDNA, was split into 1 kb windows. Within each window, the normalised ChIP read count was divided by the input read count as many regions show copy numbers deviating from one relative to the genomic annotation. For rDNA windows, only the consensus 9 kb rDNA repeat region was included.

For *Figure 1D*, RNAseq data for histone depletion was obtained from GEO (GSE29294), and mapped to the yeast genome as above. Read counts were calculated for each annotated CDS, ignoring strand as the data in GSE29294 is non-strand specific. For each CDS, change across age was calculated as the mean read count at 48 hr minus mean read count at log using the data from this work, which was plotted against the difference in read count between pre- and post-histone depletion from GSE29294.

All raw mRNAseq and ChIPseq data have been deposited at GEO under accession number GSE107744. Gene expression data underlying the hierarchical clustering diagrams and per-locus data for the ChIPseq analysis is given in *Supplementary file 3*. Additional ageing H3K4me3 ChIP data without the *Drosophila* chromatin spike-in has been deposited at GEO under accession number GSE120191.

## Acknowledgements

We thank Kristina Tabbada and Clare Murnane in the BI Next Generation Sequencing Facility for data generation, Felix Krueger, Anne Segonds Pichon and Simon Andrews of the BI Bioinformatics facility along with Mikhail Spivakov for statistical and bioinformatics advice, Gavin Kelsey for critical reading, and Dan Gottschling for the MEP strains. Funding was from the Wellcome Trust [088335,110216], the BBSRC [BI Epigenetics ISP: BBS/E/B/000C0423], and the ERC [EpiGeneSys Network]. Use of the BI Sequencing and Bioinformatics Facilities was supported by BI's BBSRC Core Capability Grant

## Additional information

### Funding

| Funder | Grant reference number | Author |
|---|---|---|
| Wellcome | 088335,110216 | Monica Della Rosa<br>Michelle King<br>Jonathan Houseley |
| Biotechnology and Biological Sciences Research Council | BBS/E/B/000C0423 | Cristina Cruz<br>Christel Krueger<br>Dorottya Horkai<br>Lucy Field |
| H2020 European Research Council | EpiGeneSys Network | Qian Gao |
| Biotechnology and Biological Sciences Research Council | BI Core Capability Grant | Jonathan Houseley |

The funders had no role in study design, data collection and interpretation, or the decision to submit the work for publication.

## Author contributions

Cristina Cruz, Formal analysis, Supervision, Investigation, Methodology, Writing—review and editing; Monica Della Rosa, Qian Gao, Dorottya Horkai, Michelle King, Lucy Field, Investigation, Methodology; Christel Krueger, Formal analysis, Visualization, Writing—review and editing; Jonathan Houseley, Conceptualization, Formal analysis, Supervision, Funding acquisition, Visualization, Methodology, Writing—original draft, Writing—review and editing

## Author ORCIDs

Cristina Cruz [ID] http://orcid.org/0000-0001-7339-5355
Christel Krueger [ID] http://orcid.org/0000-0001-5601-598X
Jonathan Houseley [ID] http://orcid.org/0000-0001-8509-1500

## Decision letter and Author response

Decision letter https://doi.org/10.7554/eLife.34081.027
Author response https://doi.org/10.7554/eLife.34081.028

# Additional files

## Supplementary files

• Supplementary file 1. Supplementary Tables S1-S7. S1: Summary of differentially expressed genes determined using DESeq2 S2: Regions of variable copy number excluded from ChIPseq analysis S3: Yeast strains used in this research S4: Oligonucleotides used in this research S5: Plasmids used in this research S6: Probes used in this research S7: Antibodies used in this research.
DOI: https://doi.org/10.7554/eLife.34081.015

• Supplementary file 2. Full results of GO analyses Full output of GO analyses presented in *Figure 1*, *Figure 1—figure supplement 1*, *Figure 2*, *Figure 2—figure supplement 1*. Only the Process Ontology is presented (as in the figures), the data here is the full output from GOrilla (cbl-gorilla.cs.technion.ac.il) with a P-value threshold of 0.001. However, only GO hits with an FDR-corrected q-value of <0.05 were considered significant in the final analysis.
DOI: https://doi.org/10.7554/eLife.34081.016

• Supplementary file 3. Data underlying figures Numerical data used to produce all figures shown. For hierarchical clustering diagrams, this includes the expression data and identifier for each gene.
DOI: https://doi.org/10.7554/eLife.34081.017

• Transparent reporting form
DOI: https://doi.org/10.7554/eLife.34081.018

## Data availability

The following datasets were generated:

| Author(s) | Year | Dataset title | Dataset URL | Database, license, and accessibility information |
| --- | --- | --- | --- | --- |
| Cristina Cruz, Monica Della Rosa, Qian Gao, Jonathan Houseley | 2017 | Characterisation of COMPASS activity in ageing yeast | http://www.ncbi.nlm.nih.gov/geo/query/acc.cgi?acc=GSE107744 | GSE107744 |
| Monica Della Rosa | 2018 | Characterisation of COMPASS activity in aged yeast | http://www.ncbi.nlm.nih.gov/geo/query/acc.cgi?acc=GSE120191 | GSE120191 |

The following previously published dataset was used:

| | | | | Database, license, and accessibility |
| --- | --- | --- | --- | --- |

| Author(s) | Year | Dataset title | Dataset URL | information |
|-----------|------|---------------|-------------|-------------|
| Gossett AJ, Lieb JD | 2012 | Effects of Histone H3 depletion on nucleosome occupancy and positioning through the S. cerevisiae genome | https://www.ncbi.nlm.nih.gov/geo/query/acc.cgi?acc=GSE29294 | GSE29294 |

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
