## [Decision Letter]

Thank you for submitting your article "Tri-methylation of Histone H3 Lysine 4 Facilitates Gene Expression in Ageing Cells" for consideration by *eLife*. Your article has been reviewed by Jessica Tyler as the Senior Editor, a Reviewing Editor, and two reviewers. The reviewers have opted to remain anonymous.

The reviewers have discussed the reviews with one another and the Reviewing Editor and feel that the following points need to be addressed prior considering the revised manuscript.

1) The reviewers requested for you and co-authors to normalize the ChIP-seq data.

2) Furthermore, it is requested that you and co-authors conduct RLS assays on H3K4 mutants.

3) Also, Western analyses on H3K4me states in the spp1 mutant during aging is also suggested by the reviewers as an essential data for revision.

We look forward to receiving your revised manuscript addressing these points. We have also included detailed reviews provided by the expert reviewers below.

Reviewer #1:

This study focuses on understanding the regulation of gene expression by tri-methylation of H3K4 during aging of budding yeast. The research described in the manuscript is worthy of praise because an extraordinary amount of work was done to obtain sufficient quantities of aged mother cells for RNAseq and ChIPseq analyses. Aging cells were obtained using the Mother Enrichment Program (MEP) system developed in the Gottschling lab. The authors verified that COMPASS mutants have significantly reduced replicative lifespans in the MEP background, results that were expected based on previous work. Analysis of RNAseq data from aging wild-type and COMPASS mutant cells by differential expression and GO analyses confirmed previous work that expression of more than one-third of annotated ORFs differ significantly between log phase cultures and aged mother cell cultures. In addition, the expression of similar categories of genes was found to be affected by aging in these and previous studies.

The authors performed differential expression analysis comparing RNAseq data from COMPASS mutants to wild-type cell data obtained at different ages in order to identify H3K4-methylation dependent genes. The results show that in log-phase cultures, a number of genes are over-expressed in COMPASS mutants when compared to wild-type cells, but this effect disappears as cells age. With age, more genes become under-expressed in COMPASS mutants. The results reveal a positive role for COMPASS in gene regulation during aging. By analyzing data from wild-type cells and COMPASS mutants during aging and comparing their data to published datasets, the authors identified nearly 300 genes whose expression is lower due to the loss of H3K4me3 during aging. The results show that COMPASS and H3K4me are required for normal levels of gene expression during aging, even though loss of COMPASS and H3K4me have relatively small effects on gene expression in log-phase cultures. Genes that are under-expressed in COMPASS mutants are required for cell wall organization and *de novo NAD+ biosynthesis*. The authors investigated the expression of several genes required for *de novo NAD+ biosynthesis* to verify a role for H3K4me in regulation of these genes.

The authors performed ChIPseq on aged cells to evaluate age-related changes in the distribution of H3K4me3 over annotated gene promoters. The ChIPseq results revealed that the peak of H3K4me3 usually associated with Pol II promoters decreases with age. Interestingly, even genes that are highly expressed in aged cells lack high levels of H3K4me3, a result that contributes to the authors' conclusion that H3K4me3 is a poor predictor of gene expression in aged cells.

All of the experiments are performed to acceptable high standards. The manuscript is well written. The discussion provides reasonable arguments that tie the new results in with previous work. The authors also discuss directions for future research.

The results of this study are likely to be of interest to scientists who study chromatin function, the regulation of gene expression, and aging.

Reviewer #2:

In this manuscript, the authors investigate the importance of H3K4me3 in regulating yeast replicative lifespan and gene expression. Replicative aging experiments were conducted on COMPASS mutants using the Mother Enrichment Program developed in the Gottschling lab. The authors show that lifespan is reduced by deletion of SWD1, which is important for all stages of H3K4 methylation, or SPP1, which has been shown to be selectively important for H3K4me3 in log phase cells. RNA-seq experiments were performed on wild type and mutant cells over a time course of aging, and the authors used these data to identify sets of genes that are either under- or over-expressed in the COMPASS mutants. The results indicate that Swd1, Spp1 and Set1 are important for normal levels of gene expression in aging cells. The primary role of Spp1, and presumably H3K4me3, appears to be as a positive regulator of genes induced upon aging. ChIP-seq experiments were performed to measure occupancy of H3K4me3 and H3 during aging. The authors conclude that H3K4me3 levels decline during aging even at genes that are upregulated. They thus argue that the expected positive correlation between H3K4me3 and gene expression is reduced during aging. While the authors have done a thorough job of analyzing gene expression during yeast replicative aging, the findings themselves, namely that mutations in the COMPASS complex affect the expression of a subset of genes, are not especially surprising. In addition, there are concerns about the way the studies were performed or interpreted that minimize the overall impact.

1) The authors have addressed the importance of H3K4me2 and H3K4me3 in replicative lifespan by measuring viability and gene expression changes in various COMPASS mutants. As they point out in their analysis of BNA gene expression levels, Set1 has been shown to target other substrates. Therefore, the changes in lifespan and gene expression as cells age may not be due to a loss of H3K4 methylation. This concern needs to be addressed. Lifespan of H3K4A or H3K4R mutants should be measured. Gene expression changes that are attributed to COMPASS during aging should be measured using the MEP method in the H3K4 mutants.

2) Many conclusions relate to the specific role of H3K4me3. As others have shown, in log phase cells, *spp1∆* leads to selective loss of H3K4me3 and retention of H3K4me2 and me1. The authors are measuring transcript levels in *spp1∆* mutants over a time course of replicative aging and are equating the effects to loss of H3K4me3 specifically. Because it's possible that the role or regulation of Spp1 changes as cells age, the authors need to confirm that H3K4me3 is lost in the *spp1∆* mutant in aging cells and that the other methyl states are retained.

3) The authors conclude that mutations in COMPASS cause "widespread gene expression defects". When they separate out the effects of histone depletion that accompany aging, they identify 204 genes that are upregulated after wild type cells are aged for 48 hours in the MEP. Of these, 13% are said to be under-expressed in a strain that should lack all H3K4 methylation (*swd1∆*). This equates to 26-27 genes and doesn't seem widespread. An explanation is needed. Also do the *spp1∆* and set1^H1017L^ mutants also affect this subset of genes?

4) The pooling together of data sets for various COMPASS mutants to improve statistical power is unconventional and reduces the information content. Why is this necessary? The authors show differential expression data for all the individual COMPASS mutants in Figure 2B,C,D, and Figure 2—Figure supplement 1. Are these data lacking in statistical significance?

5) Throughout the figures, the number of genes reported as significantly altered varies depending on the mutant, the time point, and whether genes are classified as over- or under-expressed. These numbers are given in parentheses, although I could not find a definition of this notation in the legends. The authors should better clarify their results, possibly with a table. How do the authors arrive at the number of ~4.5% of genes being impacted by COMPASS during replicative aging? Also, while this is not an insignificant number of genes, I'm not sure "widespread" accurately reflects the number of genes affected.

6) The five *spp1∆* replicates appear to have quite different levels of transcripts for the 19 overexpressed genes, raising concerns about those conclusions.

7) Figure 3 – In this figure, the authors follow up on the RNA-seq data, which indicate under-expression of BNA genes during aging in COMPASS mutants. Northern blots were done on log phase cells, not aging cells, to analyze the induction of the BNA genes in response to nicotinamide. There are several concerns with this figure. First, it's not clear how the analysis relates to BNA expression during aging. Second, the northern blots should be shown. Third, the authors conclude that induction of BNA1, BNA2, BNA4, BNA5 and BNA6 is impaired in *spp1∆* cells. From the way the data are presented, it appears that both the basal and induced levels of expression are lower in the spp1 mutant, but it's not clear that the fold induction is greatly impaired. Fold induction values should be given. Finally, because they lack all K4 methylation, the H3K4A and H3K4R mutants would be expected to phenocopy the *swd1∆* mutant for induction of the BNA genes. It is difficult to know from the presented graphs whether this is the case. It's also not clear why these mutants would phenocopy *spp1∆* and not *swd1∆*, as the authors suggest for certain BNA genes. An explanation is needed.

8) As the Tyler lab originally showed for aging cells, the method for normalizing ChIP-seq or MNase-seq data can have profound effects on interpretations when global changes in histone occupancy occur. It doesn't appear that the authors used a spike-in control to normalize their histone occupancy data or that the H3K4me3 ChIP-seq data are normalized relative to total H3. The lack of normalization to spike-ins and total histone levels confounds interpretations. Using the MEP strategy, Hu et al., (2014) reported a 50% decrease in nucleosome occupancy as cells age. The aging effect on H3 ChIP signals shown in Figure 4 and associated supplement appears much smaller than 50%. The discrepancy with the spike-in normalized data from the Tyler lab raises concerns about whether the data of Cruz et al., are properly normalized. This is a major concern as the authors’ main conclusions rest on these results.

---

## [Author Response]

The reviewers have discussed the reviews with one another and the Reviewing Editor and feel that the following points need to be addressed prior considering the revised manuscript.1) The reviewers requested for you and co-authors to normalize the ChIP-seq data.

We have set up a normalised low-cell number ChIP method using a spike-in of *Drosophila* chromatin introduced at cell lysis to allow corrections for loss of material during sample processing and differences in ChIP efficiency. We think that this provides a much more accurate measure of ChIP signals from young and aged cells. This was well worth the effort as the normalised ChIP data reveal that H3K4me3 is not lost with age but rather accumulates and is re-distributed within the genome, an observation that fits much better with the main message of the paper regarding the importance of H3K4me3 in gene expression. The new ChIP data is otherwise very similar to our previous data, with H3K4me3 showing reduced dynamic range in aged cells and reduced definition of the promoter peak due increased signal in non-promoter regions.

2) Furthermore, it is requested that you and co-authors conduct RLS assays on H3K4 mutants.

We have performed this experiment, now Figure 1—figure supplement 1C. Results are in accord with the previous mutant analysis.

3) Also, Western analysis on H3K4me states in the spp1 mutant during aging is also suggested by the reviewers as an essential data for revision.

We have performed this experiment, now Figure 2—figure supplement 1E. We show that the effect of *spp1*Δ on H3K4me3 and H3K4me2 is the same in young and aged cells.

Reviewer #2:[…] 1) The authors have addressed the importance of H3K4me2 and H3K4me3 in replicative lifespan by measuring viability and gene expression changes in various COMPASS mutants. As they point out in their analysis of BNA gene expression levels, Set1 has been shown to target other substrates. Therefore, the changes in lifespan and gene expression as cells age may not be due to a loss of H3K4 methylation. This concern needs to be addressed. Lifespan of H3K4A or H3K4R mutants should be measured. Gene expression changes that are attributed to COMPASS during aging should be measured using the MEP method in the H3K4 mutants.

We have had considerable difficulty in introducing histone point mutations into the MEP background both by sporulation and by delitto perfetto for reasons that remain unclear, as we have had no problem using either method to introduce other mutations in the same background. Nonetheless, we were able to obtain a set of haploid H3K4 mutants in the MEP background, and these show similar lifespan defects to the COMPASS mutants (Figure 1—figure supplement 1C), although we have to perform the assay at an earlier time point since the MEP is not as tight in haploids as diploids, as noted in the original description of the system (Lindstrom and Gottschling, 2009). Because of these issues, we were not able to perform gene expression studies in these mutants, but we note that in the validation of BNA gene expression, the histone point mutant phenotypes were coherent with the effects of COMPASS being exerted through H3K4 methylation.

2) Many conclusions relate to the specific role of H3K4me3. As others have shown, in log phase cells, spp1∆ leads to selective loss of H3K4me3 and retention of H3K4me2 and me1. The authors are measuring transcript levels in spp1∆ mutants over a time course of replicative aging and are equating the effects to loss of H3K4me3 specifically. Because it's possible that the role or regulation of Spp1 changes as cells age, the authors need to confirm that H3K4me3 is lost in the spp1∆ mutant in aging cells and that the other methyl states are retained.

This is a good point. We have added a Western blot for H3K4me2 and H3K4me3 in log phase and 24 hour aged cells in Figure 2—figure supplement 2E. This clearly shows that H3K4me3 is absent in *spp1*Δ cells irrespective of age, while H3K4me2 is unaffected by loss of Spp1 at either age. Both antibodies we have tried against H3K4me1 give a very low signal and did not reveal detectable bands from the small amount of protein available from an ageing purification. However, given that H3K4me2 is unaffected we consider it highly unlikely that H3K4me1 is altered.

3) The authors conclude that mutations in COMPASS cause "widespread gene expression defects". When they separate out the effects of histone depletion that accompany aging, they identify 204 genes that are upregulated after wild type cells are aged for 48 hours in the MEP. Of these, 13% are said to be under-expressed in a strain that should lack all H3K4 methylation (swd1∆). This equates to 26-27 genes and doesn't seem widespread. An explanation is needed. Also do the spp1∆ and set1^H1017L^ mutants also affect this subset of genes?

The reviewer has misunderstood our point here. From the initial tightly restricted dataset of 204 genes, we found that 13% were under-expressed in aged but not log phase cells. This observation leads us to expand this analysis to the whole genome, finding almost 500 genes to be misregulated in aged COMPASS mutants, and it is this value we describe as ‘widespread’. We have reworded for clarity. Yes, the 26 genes are also under-expressed in the spp1 and set1^H1017L^ mutants, we have noted this in the text.

4) The pooling together of data sets for various COMPASS mutants to improve statistical power is unconventional and reduces the information content. Why is this necessary? The authors show differential expression data for all the individual COMPASS mutants in Figure 2B, C, D, and Figure 2—figure supplement 1. Are these data lacking in statistical significance?

The reason for pooling the data was that producing 4-5 replicates for each time point for each mutant would have taken a prohibitive amount of time; each ageing cell purification represents a substantial investment (reviewer 1 notes that the dataset we have already obtained of aged samples for RNAseq and ChIPseq is “an extraordinary amount of work”). Although comparing wild-type to *swd1*∆ alone also provides quite a large number of significantly differentially expressed genes, the GO categories enriched proved to be less informative than those obtained through the pooled strategy. However, we take the reviewers point that this is non-standard, and we have therefore provided data comparing the wild-type to *swd1*∆ alone in Figure 2—figure supplement 1C. Because the pooled dataset was not informative about effects of H3K4me3, we had already included additional data for *spp1*∆ at log and 48 hours to allow significant differentially expressed genes to be detected for wild-type versus *spp1*∆ at these time-points (see Supplementary file 1).

Of the figures questioned, in Figures 2B and Figure 2—figure supplement 1A/B the genes shown are significantly different when comparing the wild-type replicate set to the pooled replicates of the *swd1*∆/*spp1*∆/*set1*^H1017L^. These differences are not necessarily significant comparing the wild-type to any individual mutant. In Figures 2C and 2D, the genes shown are significantly different between wild-type and *spp1*∆ at the indicated time points.

5) Throughout the figures, the number of genes reported as significantly altered varies depending on the mutant, the time point, and whether genes are classified as over- or under-expressed. These numbers are given in parentheses, although I could not find a definition of this notation in the legends. The authors should better clarify their results, possibly with a table. How do the authors arrive at the number of ~4.5% of genes being impacted by COMPASS during replicative aging? Also, while this is not an insignificant number of genes, I'm not sure "widespread" accurately reflects the number of genes affected.

The numbers given in parenthesis are simply the actual number of genes found to be differentially expressed based on a DESeq2 analysis – we have re-worded the figure legends to be clearer about precisely what is shown. We have also included a table summarising the DESeq2 data (Table S1) as the reviewer suggests. The figure of 4.5% is calculated from 297 genes being significantly under-expressed in the pooled COMPASS mutant dataset compared to the wild-type at 48 hours, relative to the 6604 ORFs annotated in the yeast genome database (297/6604*100=4.497%).

We agree that we have used the word “widespread” to represent both the upregulation of gene expression with age (which affects almost all genes) and the 4.5% of genes we detect as H3K4 dependent, and this is inconsistent. We have therefore changed the wording in the abstract (the only place we can find where this word was used in this context) to give the number of genes affected and remove the word “widespread”.

6) The five spp1∆ replicates appear to have quite different levels of transcripts for the 19 overexpressed genes, raising concerns about those conclusions.

We did not intend to suggest that the 19 overexpressed genes were consistent and agree with the reviewer that they are not. Rather, we restricted our conclusions to the under-expressed set which is highly reproducible across replicates. We have changed the wording slightly for clarity.

7) Figure 3 – In this figure, the authors follow up on the RNA-seq data, which indicate under-expression of BNA genes during aging in COMPASS mutants. Northern blots were done on log phase cells, not aging cells, to analyze the induction of the BNA genes in response to nicotinamide. There are several concerns with this figure. First, it's not clear how the analysis relates to BNA expression during aging. Second, the northern blots should be shown. Third, the authors conclude that induction of BNA1, BNA2, BNA4, BNA5 and BNA6 is impaired in spp1∆ cells. From the way the data are presented, it appears that both the basal and induced levels of expression are lower in the spp1 mutant, but it's not clear that the fold induction is greatly impaired. Fold induction values should be given. Finally, because they lack all K4 methylation, the H3K4A and H3K4R mutants would be expected to phenocopy the swd1∆ mutant for induction of the BNA genes. It is difficult to know from the presented graphs whether this is the case. It's also not clear why these mutants would phenocopy spp1∆ and not swd1∆, as the authors suggest for certain BNA genes. An explanation is needed.

First: We are slightly surprised by this concern as in Figure 3 and Figure 3—figure supplement 1 we provide side-by-side comparisons of the induction of the *BNA* genes in log phase and during ageing. Generally, we think it is major strength of our analysis that we could recapitulate the effects observed during ageing in log phase cells, as this removes concerns about indirect effects stemming from age-related physiological differences.

Second: We have included example northern blots as Figure 3—figure supplement 2.

Third: We were not claiming that induction is affected per se, rather that gene expression is affected in general, which affects both the basal and induced level, but is most easily seen as an apparent reduction in the induced level and have changed wording in a few places where this was less than clear. Of course, in many cases where we see a reduction in the induced level it is most likely a combination of effects on the basal level and the induction, but in one case (*BNA4*) induction is substantially reduced without the basal level being altered. We have now included a table of fold changes so readers can judge this for themselves in Figure 3—figure supplement 2C.

Finally: As noted in the response to reviewer 1 point 9, we agree that actually the effects of the histone mutants are equivalent to the effect of *swd1* in all cases except for *BNA1* and even here the histone mutant expression lies in between the effects seen in *swd1* and *spp1* mutants, and therefore still largely reflects the complete loss of H3K4 methylation. We have changed the text appropriately, and this certainly makes the conclusions more logical.

8) As the Tyler lab originally showed for aging cells, the method for normalizing ChIP-seq or MNase-seq data can have profound effects on interpretations when global changes in histone occupancy occur. It doesn't appear that the authors used a spike-in control to normalize their histone occupancy data or that the H3K4me3 ChIP-seq data are normalized relative to total H3. The lack of normalization to spike-ins and total histone levels confounds interpretations. Using the MEP strategy, Hu et al., (2014) reported a 50% decrease in nucleosome occupancy as cells age. The aging effect on H3 ChIP signals shown in Figure 4 and associated supplement appears much smaller than 50%. The discrepancy with the spike-in normalized data from the Tyler lab raises concerns about whether the data of Cruz et al., are properly normalized. This is a major concern as the authors’ main conclusions rest on these results.

We are grateful to the reviewer for compelling us to perform these experiments as the results are extremely informative. We have now optimised a low cell-number ChIP method for yeast including a *Drosophila* chromatin spike-in added at the point of cell lysis. This allows us to correct for loss of material during processing (considerable and variable in low cell number applications in our experience) and also for differences in ChIP efficiency.

The spike-in normalised H3K4me3 data reveals that although definition and dynamic range of promoter-associated H3K4me3 peaks are lost with age as we originally concluded, this is not due to a global loss of the mark but rather to a genome-wide gain, with additional H3K4me3 being deposited at promoters of low expressed genes but also accumulating in other regions of the genome. This observation makes much more sense in the context of the genome-wide gene induction associated with ageing. Therefore, our data still shows global changes and altered distribution of H3K4me3 with age but fits much better with the main focus of this paper, which is the importance of H3K4me3 in facilitating gene expression.

We are aware that this result is at odds with our original Western blotting data, which showed a reduction in H3K4me3 with age, and we have considered carefully the root of this discrepancy. We normalised the Western blot to total protein levels as other have previously described, however this assumes that the per-cell total protein content does not change with age and we are not sure this assumption is correct. Ribosomal protein constitutes 10-25% of total protein (Warner, 1971, Warner 1999), and is a particularly large part of the well-defined low molecular weight protein bands that are most easily used in total protein normalisation for histones. However, studies using both shotgun mass spectrometry of total protein and microfluidic analysis of a GFP-labelled ribosomal protein have concluded that ribosomal protein levels rise substantially with age (Janssens 2015, Janssens 2016). We also find by northern blotting that the total amount of rRNA rises 3-fold across 48 hours relative to *ACT1* or *PGK1* mRNA (which are stable across age in both our analysis and that of Hu et al.,); since rRNA is neither synthesised nor stable in the absence of stoichiometric ribosomal proteins, this implies a substantial increase in ribosomal protein per cell with age (see Author response image 1). This changing normalisation control would result in an apparent loss of H3K4me3 with age.

**Author response image 1. respfig1:** Ratio of rRNA to ACT1 mRNA levels in wild-type cells across age.

The reviewer’s general concern about H3 levels is not resolved by our new normalisation. Our data does show that histones redistribute to rDNA regions in aged cells rather than being lost entirely from the genome, and we note that an equivalent redistribution is detectable in the MNase-seq data of Hu et al. The dynamics of histone occupancy with age are therefore much more complex than a simple genome-wide reduction in occupancy. We further note that H3 ChIPseq and MNase-seq are not perfectly comparable and these studies utilised very different growth conditions (carbon source was raffinose/galactose in Hu et al. as opposed to glucose in our study). A detailed dissection of histone loss and re-distribution during ageing is outside the scope of this work, which is focused on H3K4 methylation, but this is certainly an important issue that we will attempt to resolve in future studies.